# ProxyThinker: Test-Time Guidance through Small Visual Reasoners

**Zilin Xiao[1], Jaywon Koo[1], Siru Ouyang[2], Jefferson Hernandez[1], Yu Meng[3], Vicente Ordonez[1]**
[1]Rice University     [2]University of Illinois Urbana-Champaign     [3]University of Virginia
{zilin,vicenteor}@rice.edu

## ABSTRACT

Recent advancements in reinforcement learning with verifiable rewards have pushed the boundaries of the visual reasoning capabilities in large vision-language models (LVLMs). However, training LVLMs with reinforcement fine-tuning (RFT) is computationally expensive, posing a significant challenge to scaling model size. In this work, we propose PROXYTHINKER, an inference-time technique that enables large models to inherit the visual reasoning capabilities from small, slow-thinking visual reasoners without any training. By subtracting the output distributions of base models from those of RFT reasoners, PROXYTHINKER modifies the decoding dynamics and successfully elicits the slow-thinking reasoning demonstrated by the emerged sophisticated behaviors such as self-verification and self-correction. PROXYTHINKER consistently boosts performance on challenging visual benchmarks on spatial, mathematical, and multi-disciplinary reasoning, enabling untuned base models to compete with the performance of their full-scale RFT counterparts. Furthermore, our implementation efficiently coordinates multiple language models with parallelism techniques and achieves faster inference compared to previous decoding-time methods, paving the way for the practical deployment of PROXYTHINKER. Code is available at https://github.com/MrZilinXiao/ProxyThinker.

## 1 INTRODUCTION

Recent advances in large language models have led to the development of systems capable of extended reasoning and deliberation, often referred to as *"slow-thinking"* models, such as OpenAI-o1 (Jaech et al., 2024), DeepSeek-R1 (Guo et al., 2025), and QwQ (Team, 2025). Unlike *"fast-thinking"* models such as GPT-4o (Hurst et al., 2024), *"slow-thinking"* models usually engage in multi-step self-reflection to produce an answer that resembles the thorough thinking process that humans make before producing a final answer for non-trivial problems. These models have achieved remarkable success in complex problem-solving benchmarks, particularly in mathematical and scientific reasoning domains (Shao et al., 2024b; Zeng et al., 2025a; Yu et al., 2025). Recent research has also extended such reflective reasoning to multimodal tasks (Huang et al., 2025; Deng et al., 2025; Yang et al., 2025; Zhou et al., 2025; Wang et al., 2025a), pushing large vision-language models (LVLMs) toward greater performance in scenarios that require structured and contextual understanding across modalities.

Many of the most effective *"slow-thinking"* models rely on reinforcement learning with verifiable rewards (RLVR) (Face, 2025; Su et al., 2025; Wei et al., 2025a), a reinforcement fine-tuning (RFT) framework that encourages the model to generate intermediate reasoning steps that lead to a correct answer for automatically verifiable tasks. While effective, this approach is computationally intensive and resource-demanding. First, the process often requires maintaining multiple model copies when using algorithms such as Proximal Policy Optimization (PPO) (Schulman et al., 2017) or Group Relative Policy Optimization (GRPO) (Shao et al., 2024a), which significantly increases memory usage. Second, the training process typically alternates between rollout and optimization phases, resulting in significant complexity and extensive training time.

Due to these high training costs, prior work has rarely applied RFT to LVLMs with more than 7 billion parameters. Recent research findings (Shah et al., 2025; Yue et al., 2025b; Gandhi et al., 2025;

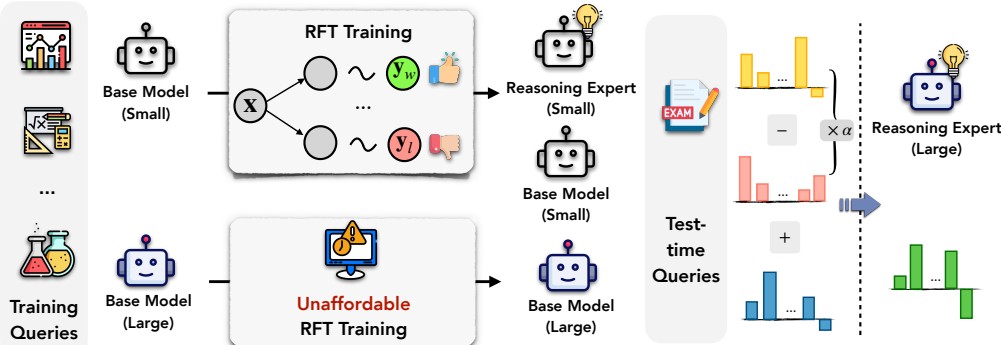

Figure 1: Illustration of PROXYTHINKER design. Without training, we migrate the reasoning behaviors learned by the small visual reasoner by capturing the token-level logit difference between a small, reasoning expert after reinforcement fine-tuning (RFT) and a small, base model. The logit difference can effectively guide a large base model to become a large reasoning expert at test time.

Liu et al., 2025a; Wang et al., 2025c) suggest that RFT does not teach new knowledge beyond the capabilities of the base model, but rather elicits and amplifies reasoning behaviors that are already included in the sampling distributions of the base model. In this work, we introduce PROXYTHINKER, a simple yet effective inference-time method that allows for efficient transfer of visual reasoning capabilities without incurring any training costs, illustrated in Figure 1. Motivated by the line of work that explores decoding-steering of language models (Liu et al., 2021; Li et al., 2023b; O'Brien & Lewis, 2023; Leng et al., 2024; Liu et al., 2024a), we propose using the difference between the last-token logits from a reasoning **Expert** model after RFT training and those from a non-RFT **Amateur** model to represent the reasoning abilities induced by RFT. Such a difference could steer a larger **Base** model toward the slow-thinking reasoning pattern. To investigate the effectiveness of PROXYTHINKER, we conduct experiments on mathematical, multi-disciplinary and complex reasoning tasks using larger base models with 32B and 72B parameters. Quantitative results show significant improvements on benchmarks such as MathVista (Lu et al., 2024), MathVerse (Zhang et al., 2024), MathVision (Wang et al., 2024a), MMMU-Pro (Yue et al., 2025a) and R1-OneVisionBench (Yang et al., 2025). For example, on the MathVision test split, we improved the accuracy of Qwen2.5-VL-32B-Instruct (Bai et al., 2025) from 38.4% to 40.8% by integrating OpenVLThinker-7B (Deng et al., 2025) as a reasoning expert, despite the latter's poor accuracy of 25.3%. This even surpasses the 40.5% achieved by the full-scale RFT model VL-Rethinker-32B (Wang et al., 2025a).

To reduce the computation overhead of running multiple models, we implement our system on top of vLLM (Kwon et al., 2023), fully exploiting modern parallelism techniques. Our optimized scheduling of logits computation across models yields a $38\times$ speedup over earlier open-sourced decoding-time steering methods. Ablation studies show that our method works robustly without any hyperparameter tuning to achieve substantial gains. We further conducted a comprehensive analysis to show emerging reasoning behaviors in PROXYTHINKER, hoping to shed light on future research work in decoding-time algorithms that enhance reasoning abilities.

## 2 METHODOLOGY

### 2.1 PRELIMINARIES

**Vision-Language Model (VLM) Decoding.** A VLM defines a conditional probability distribution $p_{\boldsymbol{\theta}}$ over output sequences, parameterized by model weights $\boldsymbol{\theta}$, and conditioned on both a textual prompt $\mathbf{x} = [x_1, \ldots, x_n]$ and a set of input images $\mathcal{I} = \{I_1, \ldots, I_k\}$. The model autoregressively generates a response $\mathbf{y} = [y_1, \ldots, y_m]$ according to:

$$p_{\boldsymbol{\theta}}(\mathbf{y} \mid \mathbf{x}, \mathcal{I}) = \prod_{j=1}^{m} p_{\boldsymbol{\theta}}(y_j \mid \mathbf{x}, \mathcal{I}, y_{<j}). \tag{1}$$

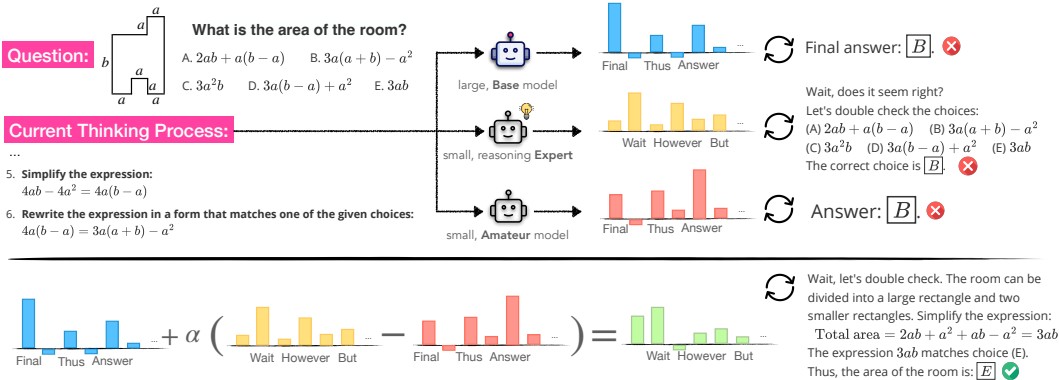

Figure 2: PROXYTHINKER with a case study from MathVision. When provided with the same incorrect thinking process, both the large and small base models tend to finalize the answer prematurely in the decoding process. The small reasoning expert shows signs of self-verification behaviors, *e.g.*, assigning high probabilities to "Wait", "However", "But". However, its limited capacity confines it to shallow reasoning, such as restating answer choices. Through logits manipulation, PROXY-THINKER transfers this reasoning behavior to the large base model, effectively triggering accurate self-correction and leading to the correct option.

**Decoding-Time Algorithm** refers to a technique that modifies a language model (LM) output distribution at inference time as a means to improve generation quality and control without training. DExperts (Liu et al., 2021) first proposes to steer the output of an LM toward desirable attributes, such as reducing toxicity and controlling sentiment, with a pair of *expert* and *anti-expert* LMs, encouraging safe continuation and penalizing toxic completions. Contrastive Decoding (Li et al., 2023b) improves open-ended text generation quality by contrasting the predictions of a large *expert* LM against those of a small *amateur* LM. The intuition behind this is that if both a big and small model are likely to produce an undesirable token (*e.g.*, a generic, repetitive word), that token score is suppressed, whereas tokens favored by the expert but not the amateur are boosted. We include additional related work in §6. In contrast to these approaches, our motivation lies in transferring the reasoning abilities of a small visual reasoner, which is orthogonal to their goals and contributions.

## 2.2 PROXYTHINKER: NEXT-TOKEN-PREDICTION WITH TEST-TIME GUIDANCE

There is increasing evidence that reinforcement fine-tuning (RFT) does not impart fundamentally new knowledge into a base model, but rather amplifies reasoning behaviors that the base model was already capable of in principle (Yue et al., 2025b). Or in other words, RFT shifts the probability mass of a model toward token sequences that exhibit structured, *"slow-thinking"* reasoning strategies, such as branching into sub-cases, backtracking after a contradiction (Swamy et al., 2025), and self-checking intermediate answers (Gandhi et al., 2025). These reasoning strategies are reflected in the high activation of relevant tokens at specific stages of the reasoning process.

In the upper part of Figure 2, we present an example from the MathVision dataset, where we provide three different vision-language models (VLMs) with the same incorrect reasoning process. Both large and small base models tend to directly provide an answer after reading the reasoning process, whereas a small RFT-trained expert exhibits reflective reasoning strategies. However, due to its limited model capacity, this reflective behavior remains *shallow* and largely restricted to restating answer options. We therefore ask: Can the reasoning skills acquired during RFT be directly transferred to a larger model via logits delta? By combining the reasoning patterns of the small RFT expert with the enhanced capacity of the large base model, we anticipate that such a transfer could deepen the model's reasoning behaviors and improve performance on reasoning-intensive tasks. In the lower part of Figure 2, we observe that applying logits delta successfully elicits the effective reflection of the large base model and ultimately leads to a correct option.

Formally, consider a pretrained VLM $\Psi$, or **Base** model, which we wish to adapt toward improved *"slow-thinking"* reasoning behavior without updating its parameters. Given a set of input images

$\mathcal{I} = \{I_1, I_2, \ldots, I_k\}$ and a text prompt $x_{<t}$ with $t$ being the current decoding time step, $\Psi$ produces a logit vector over the vocabulary, conditioned jointly on both modalities. The goal is to guide $\Psi$ to behave as if it had undergone RFT while avoiding any parameter updates to $\Psi$. To achieve this, we introduce two auxiliary models: a small, base **Amateur** model $\psi_0$ and its RFT counterpart, reasoning **Expert** model $\psi_1$, which is much cheaper to tune than tuning the large model $\Psi$ itself with the RFT method. The proposed PROXYTHINKER modifies the output distribution of $\Psi$ at inference time using a logit shift computed from the difference between the output logits of $\psi_1$ and $\psi_0$.

At each decoding step $t$, we condition $\Psi$, $\psi_1$, and $\psi_0$ on the shared image set $\mathcal{I}$ and the current text prefix $x_{<t}$ to compute logits $z_\Psi$, $z_{\psi_1}$, and $z_{\psi_0}$, respectively. A hyperparameter $\alpha \in \mathbb{R}^+$ controls the influence of this difference signal, which we will discuss its impact in §4.1. The adjusted distribution from PROXYTHINKER model $\hat{\Psi}$ is given by:

$$p\left(x_t \mid x_{<t}, \mathcal{I}\right) = \text{softmax}\left[z_\Psi\left(x_t \mid x_{<t}, \mathcal{I}\right) + \alpha \cdot \left(z_{\psi_1}\left(x_t \mid x_{<t}, \mathcal{I}\right) - z_{\psi_0}\left(x_t \mid x_{<t}, \mathcal{I}\right)\right)\right]. \quad (2)$$

A token $x_t$ is then sampled from this adjusted distribution and appended to the input sequence $x_{<t}$, forming $x_{\leq t}$, used as the next-step input for all three models — $\Psi$, $\psi_0$, and $\psi_1$ — in the subsequent decoding iteration. This feedback loop continues autoregressively until the end of the generation.

## 2.3 IMPLEMENTATION

In practice, using multiple models inevitably introduces computational overhead. Prior decoding-time algorithms have naively employed coarse-grained pipeline parallelism, where the execution of different models is sequential and synchronous. For instance, the expert model must wait for the base model to complete all pipeline stages before proceeding. This leads to a significant amount of idle time in the multi-GPU pipeline, as shown in Figure 3.

We address this by first implementing collaborative decoding with multiple models in the vLLM framework (Kwon et al., 2023), which enables us to leverage modern inference features such as KV cache, tensor parallelism, and continuous batching, corresponding to the *Ours* implementation in Figure 3. Furthermore, we

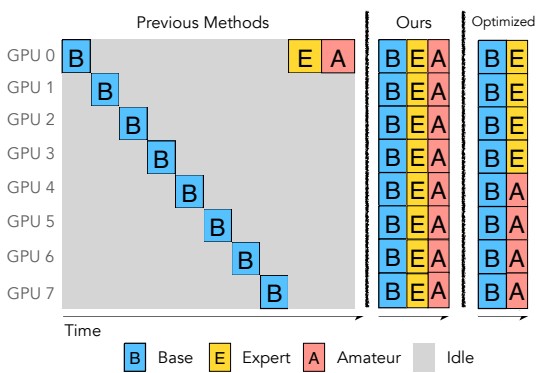

Figure 3: Comparison between PROXYTHINKER implementation and previous methods.

find that small models typically exhibit marginal returns in speedup as the tensor parallel size increases (Wu et al., 2025). Therefore, we apply tensor parallelism to the large base model and asynchronously run the small expert and amateur models in separate tensor-parallel process groups to further increase throughput. Logits from different models are synchronized via collective communication right before sampling. This optimized scheduling of multiple language models, as indicated in *Optimized* in Figure 3, minimizes GPU idle time and significantly reduces inference time, as shown in §4.4.

## 3 EXPERIMENTS

In this section, we first introduce the evaluation setup, including models and benchmarks used and evaluation strategies. We next report the effectiveness of PROXYTHINKER from these perspectives: mathematical and multi-disciplinary reasoning.

### 3.1 EXPERIMENT SETUP

**Benchmarks and Evaluation Setup.** To quantitatively assess PROXYTHINKER's reasoning capabilities across various domains, we evaluate it on several established benchmark datasets. For math-related reasoning, we use MathVerse (Zhang et al., 2024), MathVista (Lu et al., 2024), and MathVision (Wang et al., 2024a). Specifically, we adopt the `testmini` splits of MathVerse and

Table 1: Performance (Accuracy %) on mathematical and multi-disciplinary reasoning benchmarks. $\alpha$ is set to 0.5 for PROXYTHINKER methods. R1-Bench stands for R1-Onevision-Bench. Overall best PROXYTHINKER method is marked with light blue. Ceiling performance of full-scale RFT expert is highlighted with gray. $\Delta$ shows the average relative improvement in % compared to the base model. 💡 indicates an RFT reasoning expert, and 🧑‍🤝‍🧑 denotes a PROXYTHINKER variant.

| Model | Expert | Math Vista | Math Verse | Math Vision | MMMU Pro | R1-Bench | $\Delta$ |
|---|---|---|---|---|---|---|---|
| Qwen2.5-VL-7B | – | 68.2 | 46.3 | 25.1 | 36.9[†] | 32.1 | – |
| 💡 OpenVLThinker-7B | – | 70.2 | 47.9 | 25.3 | 39.4[*] | 32.9[*] | – |
| 💡 ThinkLite-VL-7B | – | 75.1 | 50.7 | 32.9 | 41.1[*] | 39.0[*] | – |
| 💡 VL-Rethinker-7B | – | 74.9 | 54.2 | 32.3 | 41.7 | 47.2[*] | – |
| Qwen2.5-VL-32B | – | 74.7 | 53.8[*] | 38.4 | 49.5[†] | 49.4[*] | 0.0 |
| 🧑‍🤝‍🧑 Qwen2.5-VL-32B | OpenVLThinker-7B | 77.4 (+2.7) | 53.8 (0.0) | **40.8** (+2.4) | 51.8 (+2.3) | **53.0** (+3.6) | +2.2 |
| 🧑‍🤝‍🧑 Qwen2.5-VL-32B | ThinkLite-VL-7B | 77.6 (+2.9) | **56.0** (+2.2) | 38.8 (+0.4) | 51.7 (+2.2) | 49.7 (+0.3) | +1.6 |
| 🧑‍🤝‍🧑 Qwen2.5-VL-32B | VL-Rethinker-7B | **78.1** (+3.4) | 55.3 (+1.5) | 39.2 (+0.8) | **52.8** (+3.3) | 52.5 (+3.1) | +2.4 |
| 💡 VL-Rethinker-32B | – | 78.8 | 56.9 | 40.5 | 50.6 | 50.8[*] | +2.4 |
| Qwen2.5-VL-72B | – | 74.8 | 55.1[*] | 38.1 | 51.6[†] | 50.4 | 0.0 |
| 🧑‍🤝‍🧑 Qwen2.5-VL-72B | OpenVLThinker-7B | 77.8 (+3.0) | 56.4 (+1.3) | 36.2 (-1.9) | 52.4 (+0.8) | 50.4 (0.0) | +0.6 |
| 🧑‍🤝‍🧑 Qwen2.5-VL-72B | ThinkLite-VL-7B | **78.7** (+3.9) | 57.2 (+2.1) | **40.4** (+2.3) | 51.7 (+0.1) | 50.2 (-0.2) | +1.6 |
| 🧑‍🤝‍🧑 Qwen2.5-VL-72B | VL-Rethinker-7B | 78.1 (+3.3) | **58.6** (+3.5) | 39.5 (+1.4) | **53.1** (+1.5) | **54.4** (+4.0) | +2.7 |
| 💡 VL-Rethinker-72B | – | 80.3 | 61.7 | 43.9 | 55.9 | 57.9[*] | +5.9 |

[†] indicates results from Wang et al. (2025a) where we adopt the same evaluation protocol.
[*] indicates reproduced results by us because the original authors did not conduct such an evaluation.

MathVista, and the full `test` split of MathVision. For assessing multi-disciplinary understanding and reasoning, we employ the MMMU-Pro (Yue et al., 2025a) overall split and the R1-Onevision-Bench (Yang et al., 2025) dataset. Following prior work, we use VLMEvalKit (Duan et al., 2024) to perform the extract-and-score procedure on MathVerse with `gpt-4o-mini` as a judge. For the remaining datasets, we apply exact matching and a rule-based grading function from their official benchmark toolkits to extract answers from model outputs and compare them to the ground truth. Unless stated otherwise, we report Pass@1 accuracy across all benchmarks using greedy decoding following previous works (Wang et al., 2025a; Deng et al., 2025) and set the hyper-parameter $\alpha = 0.5$.

**Model Selection.** PROXYTHINKER employs three distinct types of models, each serving a unique role: a large **Base** model to be steered, a compact reasoning **Expert** model, and a compact **Amateur** model. We use Qwen2.5-VL-32B-Instruct and Qwen2.5-VL-72B-Instruct (Bai et al., 2025) as Base Models, while Qwen2.5-VL-7B-Instruct serves as the Amateur Model across all RFT Experts. In later sections, PROXYTHINKER-32B and PROXYTHINKER-72B correspond to our methods using 32B and 72B base models, respectively. Based on differing training paradigms and data selection strategies, we include these public models as the compact reasoning experts: OpenVLThinker-7B (Deng et al., 2025), ThinkLite-VL-7B (Wang et al., 2025b) and VL-Rethinker-7B (Wang et al., 2025a). We refer to Appendix A.1 for details of the used benchmarks and reasoning expert models.

## 3.2 MAIN RESULTS ON MATHEMATICAL AND MULTI-DISCIPLINARY REASONING

To further investigate the generality and scalability of PROXYTHINKER, we examine whether similar effects can be observed on widely used mathematical and multi-disciplinary reasoning tasks using 32B and 72B models with different types of RFT reasoning experts. The prompt template for each reasoning expert is attached in Appendix A.3. We report these results in Table 1. To explore the upper bound of our method, we also use two larger models, VL-Rethinker-32B and VL-Rethinker-72B, which are directly trained via RFT, as ceiling performance references.

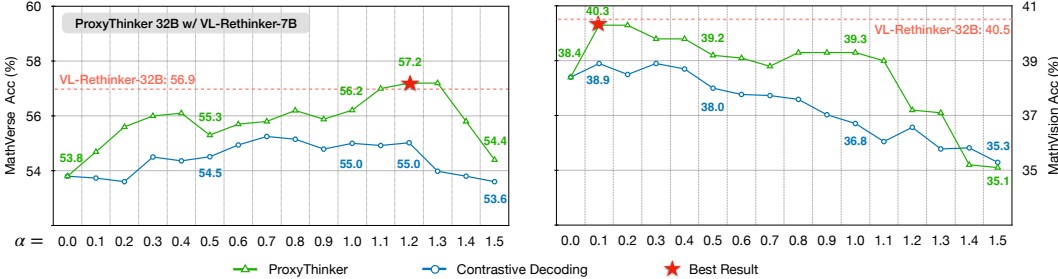

Figure 4: Hyper-parameter search results of $\alpha$ on MathVerse and MathVision benchmarks with VL-Rethinker-7B as the expert model. Red dotted lines provide the ceiling performance of VL-Rethinker-32B, a full-scale RFT reasoning expert.

**PROXYTHINKER provides consistent improvements on nearly all benchmarks.** With the exception of the MMMU validation set, we observe consistent performance improvements across all benchmark tasks and model-expert combinations. For example, using OpenVLThinker-7B as an expert improves Qwen2.5-VL-32B-Instruct's MathVision test accuracy from 38.4% to 40.8%, surpassing even the fully RFT-trained VL-Rethinker-32B (40.5%). This improvement is unlikely due to knowledge transfer, as OpenVLThinker-7B achieves only 25.3% on MathVision. Rather, it suggests that the reasoning patterns of the small expert have been effectively extracted and applied to enhance the large base model's reasoning abilities that otherwise would require full-scale RFT to activate. We discuss the domain-agnostic transferability of PROXYTHINKER in Appendix B.1, providing strong evidence that PROXYTHINKER enables the transfer of a general visual reasoning prior.

**Quality of RFT expert generally determines the degree of PROXYTHINKER improvement.** Consistent with our intuition, a stronger RFT expert tends to provide more structured reasoning paths, enhancing the base model's reasoning abilities more effectively. VL-Rethinker-7B, the most competitive of the experts, achieves the best overall results with both Qwen2.5-VL-32B and 72B. To gain deeper insights into the role of RFT experts, we discuss in Appendix B.2 on how underperforming experts may introduce negative effects on PROXYTHINKER.

**PROXYTHINKER maintains competitive scalability across model sizes.** At the 32B scale, we observe that PROXYTHINKER achieves a similar average relative improvement to full RFT models at the 32B scale. This indicates that ProxyThinker can extract and transfer structured reasoning behaviors almost as effectively as direct RFT training at this size. Notably, PROXYTHINKER can even surpass the RFT ceiling on certain tasks. At the 72B scale, ProxyThinker continues to yield consistent gains across benchmarks – smaller in magnitude but still positive – demonstrating that the approach remains effective even as the base model grows.

## 4 ANALYSIS

In this section, we further analyze PROXYTHINKER both qualitatively and quantitatively, with a focus on hyperparameter sensitivity, the type of RFT expert, the emergence of transferred reasoning behaviors, and the computational overhead of multiple model inference.

### 4.1 HYPERPARAMETER SENSITIVITY

The only hyperparameter in PROXYTHINKER, $\alpha$ in Equation 2, regulates the scaling of the logits difference applied to the base model's logits, where a smaller $\alpha$ results in the logits closer to the original base model. To validate the robustness, we run experiments with Qwen2.5-VL-32B-Instruct with VL-Rethinker-7B as the expert model, across MathVision and MathVerse benchmarks. We show a sweep of results on $\alpha \in [0.1, 1.5]$ in increments of 0.1 in Figure 4. To demonstrate the advantage of PROXYTHINKER design, we additionally include a modified contrastive decoding baseline shown as the blue curve in the sweep, which relies on the base model distribution $z_{\text{base}}$, expert distribution $z_{\text{expert}}$ and performs decoding by sampling from $z_{\text{base}} + \alpha \cdot z_{\text{expert}}$.

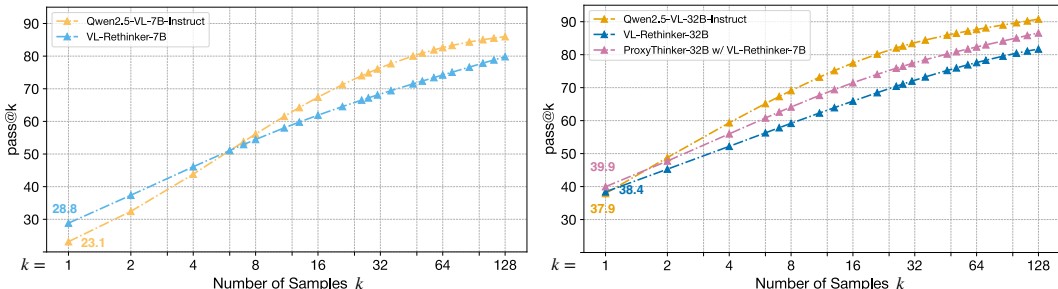

Figure 5: Pass@$k$ curve of base model, VL-Rethinker-7B RFT expert and PROXYTHINKER.

We found PROXYTHINKER to be highly robust to $\alpha$, as values between 0.1 to 1.0 could yield noticeable improvements over the base model ($\alpha = 0.0$). Specifically, we highlight that the results reported in Table 1 with the default setting of $\alpha = 0.5$ do not reflect the best performance. For VL-Rethinker-7B, the best $\alpha$ yields 40.3% on MathVision and 57.2% on MathVerse, surpassing the 39.2% and 55.1% reported in Table 1, and even reaching the ceiling performance of VL-Rethinker-32B, a full-scale, large reasoning expert. In addition, we observe that as $\alpha$ approaches 1.5, the performance gradually converges toward that of the small expert, which aligns with our intuition: a larger $\alpha$ amplifies the influence of the logits delta, resulting in the decoding process being increasingly dominated by the small reasoning expert. Finally, the performance gap between PROXYTHINKER and contrastive decoding indicates that the Amateur model in PROXYTHINKER is a critical component. It plays a key role in calibrating the logits distribution introduced by the RFT-trained small visual reasoning expert and effectively leveraging the capabilities of the large base model.

## 4.2 MEASURE THE REASONING CAPACITY BOUNDARY BY PASS@$k$

Recent studies (Yue et al., 2025b) argue that single-pass greedy decoding reflects only average-case performance of a model, and advocate for pass@$k$ (Chen et al., 2021) as a better measure of reasoning capacity boundary. These studies found that while RFT improves sampling efficiency, it lowers the reasoning boundary: as $k$ increases, RFT models produce less diverse outputs, leading to lower pass@$k$ scores compared to base models.

To investigate whether PROXYTHINKER inherits this phenomenon, we first compare the pass@$k$ curves of Qwen2.5-VL-7B-Instruct (base model) and VL-Rethinker-7B (reasoning expert) on a MathVision subset on the left of Figure 5 with evaluation details in Appendix A.2. Echoing prior findings, VL-Rethinker-7B outperforms the base model at $k \leq 4$ in pass@$k$ but is suppressed as $k$ grows, indicating a reduced reasoning boundary brought by RFT training. We then compared Qwen2.5-VL-32B-Instruct (base model), VL-Rethinker-32B (full-scale RFT expert), and ProxyThinker-32B w/ VL-Rethinker-7B shown on the right of Figure 5. ProxyThinker outperforms both at $k = 1$, though the base model dominates as $k$ increases. In terms of curve trends, the PROXYTHINKER curve consistently lies between those of the base model and the reasoning expert for $k \geq 2$, indicating that PROXYTHINKER is able to transfer the efficient sampling behavior of the expert while partially retaining the diverse exploration capability of the base model.

## 4.3 EMERGING REASONING BEHAVIORS

In addition to the example in Figure 2, we present a qualitative MathVision test case in Figure 7, showcasing the reasoning process of three models to examine the emerging reasoning abilities of PROXYTHINKER. We find that the large base model makes a commonsense error and fails to reach the correct answer, while the small reasoning expert remains confined to shallow reasoning patterns. In contrast,

Table 2: Reasoning pattern statistics on MathVision of Base, Expert, and PROXYTHINKER with relative gains over Base. Base model is Qwen2.5-VL-32B and Expert model is OpenVLThinker-7B.

| Model | Backtracking | Verification | Subgoal |
|---|---|---|---|
| Base | 203 | 1206 | 2980 |
| Expert | 805 (+296.6%) | 1585 (+31.4%) | 2427 (-18.5%) |
| ProxyThinker | 482 (+137.4%) | 1736 (+43.9%) | 2998 (+0.6%) |

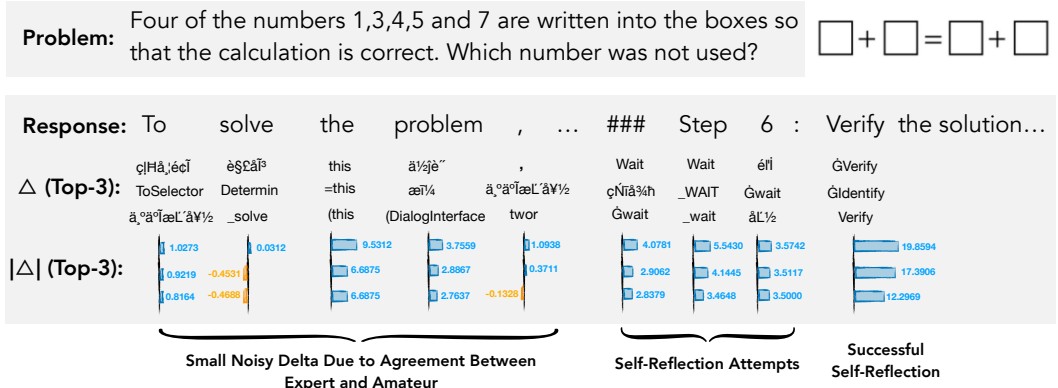

Figure 6: Case study presenting the top-3 tokens of $\Delta$ between small reasoning expert (OpenVLThinker-7B) and amateur model (Qwen2.5-VL-7B) per decoding step. At early decoding steps, $\Delta$ tends to be noisy and creates a minor shift over byte-level tokens (shown in those random non-ASCII characters) in the vocabulary. In later decoding steps, $\Delta$ becomes structured and steers the response to self-reflection patterns. $|\Delta|$ indicates the magnitude of top-3 logits.

PROXYTHINKER successfully combines the strengths of both: it inherits the explicit "thinking process" tags (`<think>`) from the small reasoning expert, while adopting the step-by-step reasoning style and even providing an interpretable, visualized solution.

To quantitatively analyze emerging reasoning behaviors, we report statistics on 3,040 MathVision samples in Table 2 on different reasoning patterns, such as backtracking, verification and sub-goal branching. Motivated by Gandhi et al. (2025), we employ `gpt-4o-mini` as a judge to classify reasoning trajectories from the large base model, the small reasoning expert, and PROXYTHINKER itself on MathVision 3040 samples. We observe that while large base models exhibit subgoal-oriented reasoning patterns, their performance on backtracking and verification remains limited. In contrast, small visual reasoners trained with RFT show significant improvements in these aspects – a strength that ProxyThinker successfully inherits. Notably, ProxyThinker also retains the subgoal planning ability, which appears to diminish in small RFT models.

## 4.4 INFERENCE OVERHEAD

Running multiple language models at inference time inevitably incurs overhead, which has been particularly problematic in previous decoding-time algorithms because of their naive implementation mentioned in §2.2. To investigate the efficiency of PROXYTHINKER, Table 3 reports runtime and accuracy on MathVision `testmini` split using Qwen2.5-VL-32B-Instruct as the base model and VL-Rethinker-7B as the expert model under several implementation variants. We also

Table 3: Inference Statistics of PROXYTHINKER.

| Method | Duration (s) | Acc (%) |
|---|---|---|
| PROXYTHINKER | | |
| Huggingface | 19133 | 40.78 |
| Ours Implementation | 578 | 41.44 |
| Optimized TP | 501 | 41.44 |
| Full-scale RFT expert | | |
| VL-Rethinker-32B | 451 | 41.77 |

ran the full-scale reasoning RFT expert with vLLM as a reference. All evaluations were conducted on 8 A100 PCIe 40GB GPUs. The results show that our implementation delivers close to 33× speedup over the Huggingface implementation, which was adopted by all previous decoding-steering methods. With the optimized tensor parallel group design, the runtime duration could be cut by another 13%, yielding a total speedup of 38×, which closely matches the duration of directly running a 32B RFT expert. We discuss the computational overhead in depth in Appendix C to demonstrate PROXYTHINKER is a practical solution in terms of memory and compute.

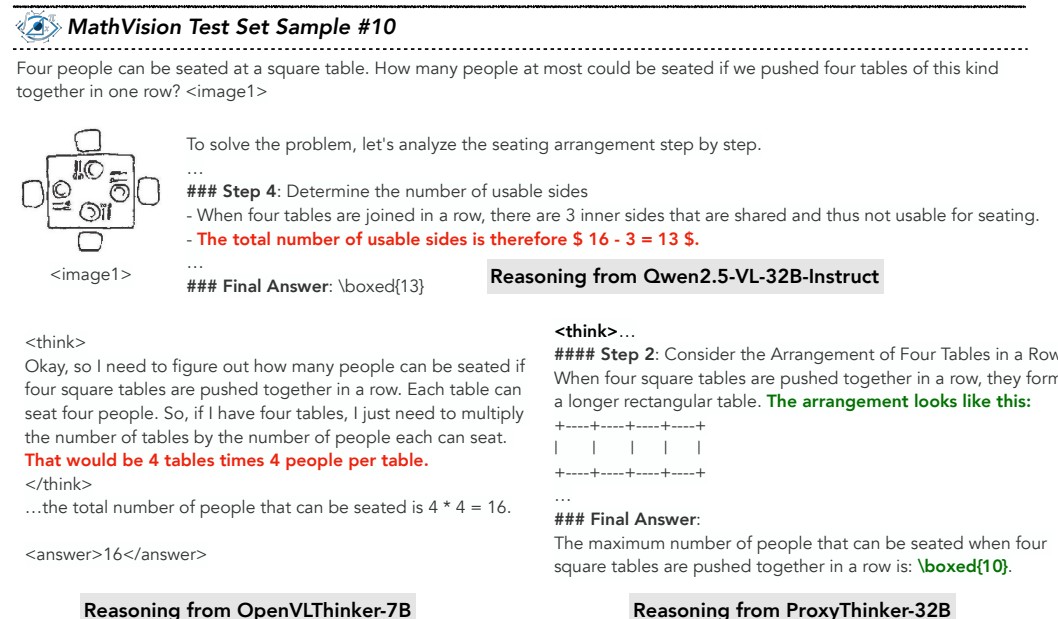

Figure 7: Case study presenting the reasoning process of large base model (Qwen2.5-VL-32B-Instruct), small reasoning expert (OpenVLThinker-7B), and PROXYTHINKER. **Bold** in the reasoning process highlights typical reasoning patterns. **Red bold** denotes wrong intermediate reasoning steps that ultimately lead to a wrong prediction, while **green bold** indicates correct reasoning steps.

## 5 CASE STUDY: TEMPORAL ANALYSIS OF THE LOGIT DELTA $\Delta$

To better understand the mechanism of how the logit delta $\Delta = z_{\text{expert}} - z_{\text{amateur}}$ affects the model behaviour, we perform a qualitative temporal analysis of $\Delta$ along a single reasoning process from PROXYTHINKER, shown in Figure 6. For each decoding step $t$, we compute $\Delta$, sort its components, and visualize the top-3 tokens together with their absolute magnitudes $|\Delta|$.

In the early part of the trajectory (the tokens "To solve the problem, ..."), the expert and base model make very similar predictions: the top-3 tokens under $\Delta$ are mostly generic or even uninterpretable, and the corresponding bars for $|\Delta|$ are close to zero. This indicates that PROXYTHINKER leaves the large model's distribution essentially unchanged when the two smaller models produce aligned logits. As the reasoning progresses, the structure of $\Delta_t$ becomes more pronounced: the highest-scoring tokens are variations of *Wait*, with noticeably larger $|\Delta|$. Here, the expert initiates an internal self-reflection step. In the final segment, the magnitude of $\Delta_t$ increases sharply and the dominant tokens shift to *Verify*, *Identify*, and similar verbs, corresponding to an explicit verification sub-goal. These dynamics show that $\Delta$ is not a static list of "reasoning tokens", but a context-dependent steering signal whose direction changes over the course of the reasoning trace.

## 6 RELATED WORK

**Large Vision-Language Models (LVLMs).** Large Vision-Language Models (LVLMs) (Wang et al., 2024b; Bai et al., 2025; Chen et al., 2024a; Zhu et al., 2024; Li et al., 2023a; Ye et al., 2023; Awadalla et al., 2023; Chen et al., 2024b) have demonstrated significant capabilities in multimodal understanding through a series of architectural advances. The Qwen2-VL (Wang et al., 2024b) and its extension introduced efficient visual tokenization through dynamic resolution mechanisms. Despite these advances, current LVLMs continue to struggle with complex reasoning tasks requiring spatial understanding and mathematical reasoning, highlighting the need for novel approaches to enhance reasoning capabilities without increasing computational demands.

**Improving Visual Reasoning with Reinforcement Learning.** Reinforcement learning has become an approach to enhance reasoning capabilities in LLMs and VLMs. OpenVLThinker-7B (Deng et al., 2025) demonstrates how reasoning capabilities can be integrated into LVLMs through an iterative self-improvement process combining supervised fine-tuning and reinforcement learning. The R1-style frameworks (Wang et al., 2025a;b; Huang et al., 2025; Zhang et al., 2025; Shen et al., 2025a; Li et al., 2025; Wei et al., 2025c) apply Group Relative Policy Optimization (GRPO) to the visual domain, demonstrating that slow-thinking reasoning abilities can be transferred to multimodal contexts. Variants like Visual-RFTs (Liu et al., 2025b) design task-specific reward and VisualThinker-R1-Zero (Zhou et al., 2025) explore applying the "Aha moment" of DeepSeek-R1 (Guo et al., 2025) to visual reasoning without supervised fine-tuning. Similar work (Wei et al., 2025b) made successful attempts in transferring reasoning behaviours from LLMs to VLMs at training time. And variations (Shen et al., 2025b; Yao et al., 2025) of policy optimization method have been proposed in post-training literatures for visual reasoning. While these approaches yield notable improvements in visual reasoning benchmarks, they require substantial computational resources, making it harder to scale. In contrast, PROXYTHINKER is a test-time decoding method that requires no further training and adds minimal computation, allowing larger VLMs to gain reasoning skills efficiently.

**Decoding-time Algorithms for Language Model.** Decoding-time algorithms steer LLMs at inference without expensive retraining. Contrastive decoding method (O'Brien & Lewis, 2023) has been applied to maximize the differences between the likelihoods of expert and amateur models to improve reasoning. CAL (Xiao et al., 2024) and VCD (Leng et al., 2024) extend this approach to multimodal settings by subtracting logits from original and perturbed images. DoLa (Chuang et al., 2024) targets the pervasive issue of LLM hallucinations – generating text not supported by factual knowledge. RAST (Ouyang et al., 2025) studies the reasoning transfer in the context of text-only tasks. DeRa (Liu et al., 2024b) and MOD (Shi et al., 2024) approximate geometric mixtures of aligned and base models or linearly combine expert predictions across objectives. In the line of these decoding-time methods, PROXYTHINKER leverages the difference between last-token logits of a reward-aligned RFT expert and a non-RFT base model, eliciting the reasoning ability learned during RFT with no additional training. A similar training-free technique was proposed recently to improve visual reasoning through the lens of model merging (Chen et al., 2025b).

## 7 CONCLUSION

We present PROXYTHINKER, a simple yet effective decoding-time algorithm for transferring visual reasoning capabilities from small visual reasoning models. PROXYTHINKER leverages the token-level logits difference between an RFT expert and an amateur model to effectively steer a large model's generation toward *"slow-thinking"*, multi-step reasoning behaviors. Through extensive experiments on vision-centric and multimodal reasoning tasks, we demonstrate that PROXYTHINKER can consistently enhance performance across model sizes, including substantial improvements on spatial, mathematical, and multi-disciplinary reasoning benchmarks. We believe PROXYTHINKER provides a promising direction for efficient reasoning transfer in large vision-language models and offers insights into the understanding of how RFT influences model behavior.

**Acknowledgements.** This work was supported by an NSF CAREER Award No. 2201710, by the Ken Kennedy Institute at Rice University, and in part by NSF Campus Cyberinfrastructure grant NSF OAC-2019007, and by Rice University's Center for Research Computing (CRC). We thank the anonymous reviewers for their constructive feedback.

## REPRODUCIBILITY STATEMENT

Experimental setups, including benchmarks, evaluation protocols, and model roles (Base / Expert / Amateur), are detailed in §3.1. Main results across mathematical and multi-disciplinary reasoning appear in Table 1 with analysis in §3.2, while efficiency considerations and multi-model scheduling are shown in Fig. 3 and summarized in Table 3. Additional implementation details, benchmark/model descriptions, and prompts are provided in Appendix A.1 and A.3. Pass@$k$ evaluation details are in Appendix A.2. A repository with inference scripts, evaluation harnesses, configuration files, and implementation is included in the abstract to reproduce Tables 1, 3, and Figures 1-5.

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

## A  IMPLEMENTATION DETAILS

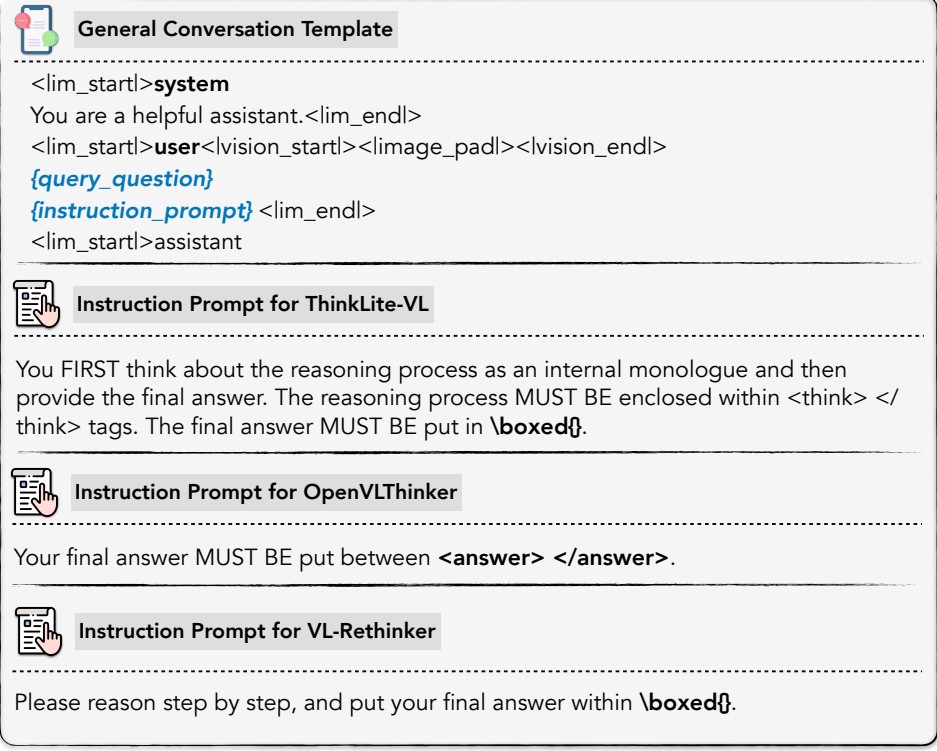

**General Conversation Template**

<|im_start|>**system**
You are a helpful assistant.<|im_end|>
<|im_start|>**user**<|vision_start|><|image_pad|><|vision_end|>
*{query_question}*
*{instruction_prompt}* <|im_end|>
<|im_start|>assistant

**Instruction Prompt for ThinkLite-VL**

You FIRST think about the reasoning process as an internal monologue and then provide the final answer. The reasoning process MUST BE enclosed within <think> </think> tags. The final answer MUST BE put in **\boxed{}**.

**Instruction Prompt for OpenVLThinker**

Your final answer MUST BE put between **<answer> </answer>**.

**Instruction Prompt for VL-Rethinker**

Please reason step by step, and put your final answer within **\boxed{}**.

Figure 8: Prompt templates of different RFT experts in §3.2.

### A.1  INTRODUCTION FOR BENCHMARKS AND REASONING EXPERT MODELS

To rigorously evaluate multimodal reasoning, we adopt several recent benchmark datasets designed to probe different dimensions of model understanding.

MathVerse (Zhang et al., 2024) contains 3,940 samples (testmini split) and tests a model's ability to interpret diagrams and solve multi-subject visual math problems presented in varying multimodal formats.

MathVista (Lu et al., 2024) provides 1,000 examples (testmini split) of visual mathematical reasoning tasks drawn from real-world diagrams, charts, and images, with both open-ended and multiple-choice formats.

MathVision (Wang et al., 2024a) includes 3,040 visual math problems across 16 mathematical disciplines, sourced from real competition problems and annotated at five distinct difficulty levels.

MMMU (Yue et al., 2024) supplies a validation split of 900 samples covering a broad range of college-level disciplines, designed to assess deliberate, multi-disciplinary multimodal understanding.

MMMU-Pro (Yue et al., 2025a) comprises 5,190 items across a Standard setting (4- and 10-option MC) and a Vision-only setting where questions are embedded directly in images, designed to stress integrated visual–textual understanding across diverse academic subjects by filtering out text-only-solvable items and augmenting candidate options.

EMMA (Hao et al., 2025) offers 2,788 multimodal problems spanning math, physics, chemistry, and coding, targeting organic cross-modal reasoning that cannot be solved by treating modalities independently.

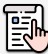 **MathVerse Answer Extraction Prompt**

I am providing you a response from a model to a math problem, termed 'Model Response'. You should extract the answer from the response as 'Extracted Answer'. Directly output the extracted answer with no explanation.\n\n

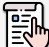 **MathVerse Answer Score Prompt**

Below are two answers to a math question. Question is [Question], [Standard Answer] is the standard answer to the question, and [Model_answer] is the answer extracted from a model's output to this question. Determine whether these two answers are consistent.
Please note that only when the [Model_answer] completely matches the [Standard Answer] means they are consistent. For non-multiple-choice questions, if the meaning is expressed in the same way, it is also considered consistent, for example, 0.5m and 50cm.
If they are consistent, Judement is 1; if they are different, Judement is 0.

Figure 9: MathVerse extraction and scoring prompt for `gpt-4o-mini` as a judge.

R1-Onevision-Bench (Yang et al., 2025) consists of 942 samples organized into five academic domains and five difficulty levels, offering a structured framework to benchmark model performance across educational subject areas.

In addition, we choose the following public models as reasoning experts based on differing training paradigms and data selection strategies:

- OpenVLThinker-7B (Deng et al., 2025): Enhances reasoning through iterative alternation between SFT and RFT stages using progressively challenging questions.
- ThinkLite-VL-7B (Wang et al., 2025b): Improves training efficiency via Monte Carlo Tree Search (MCTS)-guided data selection to achieve data-efficient RFT paradigm.
- VL-Rethinker-7B (Wang et al., 2025a): Addresses the vanishing advantage issue through Selective Sample Replay and enforces self-reflection using the Forced Rethinking technique.

## A.2 PASS@$k$ EVALUATION DETAILS

For the pass@$k$ evaluation in §4.2, we use a temperature of 0.6 and a top-p value of 0.95, allowing a maximum generation of 4,096 tokens during generation. To prevent the model from randomly guessing the multiple-choice answer through repeated sampling, we filtered the MathVision dataset to a subset where no multiple-choice problems are included, resulting in 114 samples.

We adopt an extended version of the pass@$k$ metric originally proposed for code generation by (Chen et al., 2021). Given a dataset $\mathcal{D}$ and a particular problem $x_i \in \mathcal{D}$, we generate $n$ independent outputs from the model, where $n \geq k$, and record $c_i$ as the number of outputs that pass the mathematical checker. The pass@$k$ score is then estimated using the following unbiased formula:

$$\text{pass @} k = \mathbb{E}_{x_i \sim \mathcal{D}} \left[ 1 - \frac{\binom{n-c_i}{k}}{\binom{n}{k}} \right] \tag{3}$$

This estimator provides a low-variance measurement of the proportion of problems solved correctly within $k$ attempts, applicable for all $k \leq n$.

---

⚛ *MMMU Validation Set Sample #898*

---

Some controversial groups that may be mislabelled as <image 1> include:

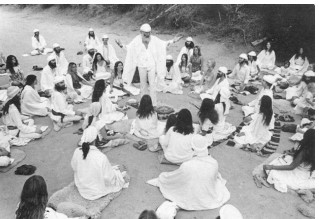

<image 1>

| Qwen2.5-VL-32B-Instruct | VL-Rethinker-7B | ProxyThinker-32B |
|---|---|---|
| To determine the correct answer, let's analyze the image and the options provided: | To determine the correct answer, let's analyze the image and the options provided: | To determine the correct answer, let's analyze the image and the options provided: |
| 1. Image Analysis:
  - The image shows a group of people sitting in a circle, dressed in white clothing, which is often associated with certain spiritual or religious groups. | 1. Image Analysis: The image shows a group of people gathered in a circle, with one individual standing in the center, possibly leading a meditation or spiritual practice. The attire and the setting suggest a spiritual or communal gathering, which is characteristic of certain New Age or spiritual movements. | 1. Image Analysis:
  - The image shows a group of people sitting in a circle, dressed in white, with one person standing in the center. This setup is reminiscent of a spiritual or religious gathering. |
| 2. Option Analysis:
  - A. Scientology and the Hare Krishna:
…
  - The image **could be consistent with** the Hare Krishna movement, as they often gather in circles for kirtan (chanting) and meditation.
…
Given the analysis, the group in the image is most consistent with the Hare Krishna movement, which is often mislabelled or associated with controversial groups. | 2. Option Analysis:
  - A. Scientology and the Hare Krishna:
…**Neither fits the description** of the image well.
  - B. the Peoples Temple and Heavens Gate:
…**This description aligns more closely** with the image.
…
Therefore, the correct answer is **\boxed{B}**. | 2. Option Analysis:
  - A. Scientology and the Hare Krishna:
… However, **they are not typically associated with**…
  - B. the Peoples Temple and Heaven's Gate: …**The image could be associated** with such groups due to the organized and somewhat cult-like appearance…
…
Given the analysis, the group in the image is most closely associated with the controversial groups known for their cult-like behavior and tragic endings. |
| **\boxed{A}** | | **\boxed{B}** |

Figure 10: A knowledge-intensive test case from the MMMU Val set with reasoning trajectories from Qwen2.5-VL-32B-Instruct, VL-Rethinker-7B and PROXYTHINKER-32B.

## A.3   PROMPT TEMPLATES

We provide the prompt templates of different reasoning experts in Figure 8. For MathVerse evaluation, we employ the prompt template in Figure 9 to first extract the answer and then score the answer using `gpt-4o-mini` as a judge, following VLMEvalKit (Duan et al., 2024).

## A.4   FAILURE CASE ANALYSIS

Although PROXYTHINKER achieves consistent improvements across multiple reasoning-intensive datasets, we observe that it struggles to deliver statistically significant gains on certain knowledge-intensive benchmarks, such as MMMU (Yue et al., 2024) – a limitation also present in full-scale RFT experts. To illustrate this, we present a test case from the MMMU Val set in Figure 10, comparing the reasoning processes of the large base model (Qwen2.5-VL-32B-Instruct), the small reasoning expert (VL-Rethinker-7B), and PROXYTHINKER. The results show that the small reasoning expert fails to accurately validate the knowledge content of answer choices, likely due to its limited model capacity. This type of knowledge verification is particularly challenging to learn via reinforcement learning with verifiable rewards. As a result, ProxyThinker inherits this limitation as well.

# B    ADDITIONAL ANALYSIS

## B.1    DOMAIN-AGNOSTIC REASONING ACTIVATION

Table 4: Performance (Accuracy %) on EMMA and R1-Bench benchmarks. Best scores are in **bold**.

| Model | Expert | EMMA Overall | R1-Bench | | | | |
|-------|--------|--------------|----------|------|---------|---------|-----------|
| | | | Overall | Math | Physics | Biology | Chemistry |
| Qwen2.5-VL-7B | – | 21.5 | 32.1 | 31.2 | 31.3 | 52.2 | 43.8 |
| OpenVLThinker-7B | – | 24.9 | 32.9 | 29.4 | 34.2 | 50.0 | 35.2 |
| Qwen2.5-VL-32B | – | 31.1 | 49.4 | 41.9 | 51.1 | 61.9 | 61.0 |
| Qwen2.5-VL-32B | OpenVLThinker-7B | **35.3** | **53.0** | **46.2** | **55.4** | **69.4** | **62.9** |

To further substantiate the claim that PROXYTHINKER reasoning transferability is domain-agnostic, we provide a detailed breakdown of its performance on EMMA and R1-OneVision-Bench (R1-Bench). The latter one covers reasoning-intensive tasks across four distinct disciplines: *Math*, *Physics*, *Biology*, and *Chemistry*. Table 4 reports the per-domain accuracies for the base model, the expert model and PROXYTHINKER.

OpenVLThinker-7B, despite being a relatively under-aligned visual reasoner, exhibits only marginal gains over the smaller Qwen2.5-VL-7B baseline, with improvements **only** in the Physics subject. However, when PROXYTHINKER is applied, we observe consistent improvements across every domain. This validates that ProxyThinker's transfer ability is not restricted to a specific subject but instead activates *domain-agnostic reasoning capabilities*. Such findings provide concrete evidence that PROXYTHINKER induces structural shifts in the output distribution of experts, enabling systematic cross-domain gains.

## B.2    QUALITY OF RFT EXPERT

Table 5: Impact of expert quality on PROXYTHINKER. Performance (Accuracy %) across three mathematical reasoning benchmarks.

| Model | Expert | Amateur | MathVista | MathVerse | MathVision |
|-------|--------|---------|-----------|-----------|------------|
| Qwen2.5-VL-32B | – | – | 74.7 | 53.8 | 38.4 |
| VL-Rethinker-7B | – | – | 74.9 | 54.2 | 32.3 |
| Qwen2.5-VL-32B | VL-Rethinker-7B | Qwen2.5-VL-7B | **78.1** | **55.1** | **39.2** |
| VLAA-Thinker-3B | – | – | 61.0 | 36.4 | 24.4 |
| Qwen2.5-VL-32B | VLAA-Thinker-3B | Qwen2.5-VL-3B | 51.5 | 49.2 | 37.8 |

To gain deeper insights into the role of RFT experts, we further examine how under-performing experts may introduce negative effects on PROXYTHINKER. In our main experiments, we primarily choose 32B/72B models as the large base and 7B models as reasoning experts, since 7B is generally the smallest scale capable of supporting effective RFT-based visual reasoning. Smaller models tend to struggle with capturing consistent reasoning patterns, leading to suboptimal transfer in the PROXYTHINKER setting.

Table 5 reports results with VLAA-Thinker-3B (Chen et al., 2025a), an early visual reasoning expert. Using VLAA-Thinker-3B as an expert not only fails to improve performance but also significantly degrades results on *MathVista* (from 61.0 to 51.5), yielding performance even worse than the 3B expert alone. This demonstrates that the quality and scale of the RFT expert are crucial: a poorly aligned or undersized expert can misguide the base model and introduce negative transfer effects.

## B.3    LANGUAGE-ONLY BENCHMARKS

We additionally evaluate PROXYTHINKER in a language-only setting in Table 6 to verify that the method is not vision-specific. All RFT reasoning models used as expert model are taken from

Table 6: Performance (Accuracy %) on language-only mathematical reasoning benchmarks. ProxyThinker is applied to base LLMs using small RFT reasoning experts as proxies. All RL models are taken from SimpleRL-Zoo (Zeng et al., 2025b). Δ shows the average *relative* improvement in % compared to the base model. Full-scale RFT expert is highlighted with gray.

| Model | Expert | Math500 | AIME'24 | GSM8K | Δ |
|---|---|---|---|---|---|
| Qwen2.5-1.5B-RL | – | 9.2 | 0.0 | 28.4 | – |
| Qwen2.5-7B-RL | – | 74.4 | 10.0 | 90.9 | – |
| Llama3.1-8B-RL | – | 9.4 | 0.0 | 13.3 | – |
| Llama3.1 70B | – | 22.2 | 3.3 | 42.4 | 0.0 |
| Llama3.1 70B + ProxyThinker-8B | Llama3.1-8B-RL | 27.0 (+4.8) | 3.3 (0.0) | 52.9 (+10.5) | +5.1 |
| Qwen2.5-14B | – | 66.4 | 3.3 | 90.4 | 0.0 |
| Qwen2.5-14B + ProxyThinker-1.5B | Qwen2.5-1.5B-RL | 73.4 (+7.0) | 10.0 (+6.7) | 92.9 (+2.5) | +5.4 |
| Qwen2.5-14B-RL | – | 78.4 (+12.0) | 10.0 (+6.7) | 94.6 (+4.2) | +7.6 |
| Qwen2.5-32B | – | 68.8 | 6.7 | 92.0 | 0.0 |
| Qwen2.5-32B + ProxyThinker-1.5B | Qwen2.5-1.5B-RL | 75.2 (+6.4) | 13.3 (+6.6) | 94.2 (+2.2) | +5.1 |
| Qwen2.5-32B + ProxyThinker-7B | Qwen2.5-7B-RL | 81.6 (+12.8) | 16.7 (+10.0) | 94.7 (+2.7) | +8.5 |
| Qwen2.5-32B-RL | – | 81.8 (+13.0) | 16.7 (+10.0) | 95.5 (+3.5) | +8.8 |

SimpleRL-Zoo (Zeng et al., 2025b). The decoding setup strictly follows §3.1 in the main paper. We select the Math500 (Lightman et al., 2023), AIME24 (Art of Problem Solving) and GSM8K (Cobbe et al., 2021) test split as preliminary testbeds to evaluate on.

We find that PROXYTHINKER yields consistent gains on all base language models and benchmarks. For Llama 3.1 70B model (Grattafiori et al., 2024), we observe noticable gains on Math500 and GSM8K, resulting in a solid average relative gain. This also indicates that PROXYTHINKER works beyond the Qwen model series. For Qwen2.5-32B model, we draw another observation which can not be seen in VLM setting due to the absence of effective small expert models – upgrading the expert from 1.5B to 7B further boosts overall performance to essentially match the fully RL-fine-tuned 32B ceiling. This echoes the conclusion we have in the main paper: a stronger RFT expert tends to provide more structured reasoning paths, enhancing the base model's reasoning abilities more effectively. Overall, these results confirm that PROXYTHINKER is a generic method that is effective beyond VLM.

## C  COMPUTATIONAL OVERHEAD ANALYSIS

**Setup and assumptions.**  We analyze the computational overhead of PROXYTHINKER by comparing the training FLOPs of small vs. large visual reasoners and the additional inference FLOPs introduced by the collaborative decoding with Amateur and Expert models. Following the reports of VL-Rethinker (Wang et al., 2025a), training VL-Rethinker-7B uses $2 \times 8$ A800 (80GB) GPUs for 8 hours, while VL-Rethinker-32B uses $10 \times 8$ A800 (80GB) GPUs for 60 hours. We assume the peak bf16 throughput of an A800 (80GB) GPU to be 312 TFLOPS and a conservative training-time FLOPs utilization of 0.5 to account for communication and system overheads.

**Training FLOPs.**  Under the above assumptions, the training FLOPs are

$$7\text{B:}\quad 16\ (\#\text{GPUs}) \times 28{,}800\text{ s} \times 3.12 \times 10^{14}\text{ FLOPS} \times 0.5 = 7.19 \times 10^{19}\text{ FLOPs,}$$

$$32\text{B:}\quad 80\ (\#\text{GPUs}) \times 216{,}000\text{ s} \times 3.12 \times 10^{14}\text{ FLOPS} \times 0.5 = 2.70 \times 10^{21}\text{ FLOPs.}$$

Hence, training a 32B reasoner costs approximately

$$2.70 \times 10^{21} - 7.19 \times 10^{19} \approx 2.62 \times 10^{21}\text{ FLOPs}$$

more than training a 7B reasoner.

**Inference FLOPs for collaborative decoding.** According to scaling-law estimates (Kaplan et al., 2020), the forward pass of a Transformer requires roughly $2\times$ (number of parameters) FLOPs per generated token.[1] Therefore, the per-token decoding cost of two 7B models (Amateur and Expert) is

$$2 \times 2 \times 7 \times 10^9 \;=\; 2.8 \times 10^{10} \text{ FLOPs/token.}$$

Let $\Delta_{\text{train}} \approx 2.62 \times 10^{21}$ be the extra training FLOPs of 32B over 7B. The number of tokens $T$ after which the accumulated collaborative-decoding overhead matches $\Delta_{\text{train}}$ satisfies

$$T \;\approx\; \frac{\Delta_{\text{train}}}{2.8 \times 10^{10}} \;\approx\; 9.37 \times 10^{10} \text{ tokens.}$$

Thus, in terms of FLOPs, the additional inference overhead of PROXYTHINKER-32B does not outweigh the training cost until roughly $9.37 \times 10^{10}$ generated tokens, *i.e.* around 93 billion tokens. Importantly, PROXYTHINKER-32B and VL-Rethinker-32B achieve comparable accuracy, while the former avoids the full training budget of a 32B reasoner.

**Latency in practice.** FLOPs do not map one-to-one to wall-clock latency. With our customized vLLM-based implementation for multi-model collaborative decoding, Table 3 shows that ProxyThinker-32B incurs only an $\sim 11\%$ latency overhead on the same hardware (from $451\,\text{s}$ to $501\,\text{s}$), indicating that the practical runtime penalty is small relative to the savings from avoiding 32B training.

**Memory overhead.** As a multi-model collaborative decoding approach, PROXYTHINKER entails additional memory to host multiple model weights. Using DeepSpeed (Rasley et al., 2020) Memory Profiler, we measure the GPU memory required to load `bfloat16` weights:

| Model Size | 7B | 32B | 72B | ProxyThinker-32B | ProxyThinker-72B |
|---|---|---|---|---|---|
| Memory $\Delta$ (GiB) | 15.31 | 64.44 | 138.86 | 95.14 (+47.6%) | 169.48 (+22.0%) |

The overhead introduced by PROXYTHINKER is approximately fixed in absolute terms, hence its *relative* percentage decreases with the base model size (e.g., $+22\%$ at 72B). This is negligible compared with the resources needed to fully train an RFT expert; *e.g.*, EasyR1 (Zheng et al., 2025) estimates that training a 32B RFT model with AMP requires at least $16 \times 80\text{GB}$ GPUs ($\approx 960\,\text{GB}$ of device memory). Moreover, modern inference frameworks typically reserve most GPU memory for KV caches, so the extra weight memory is less constraining in practice. Prior collaborative-decoding or multi-model methods (*e.g.*, contrastive decoding (Li et al., 2023b), ProxyTuning (Liu et al., 2024a)) exhibit similar memory characteristics.

**Takeaways and limitations.** PROXYTHINKER is an inference-time strategy that trades modest multi-model decoding overhead for avoiding the substantial training cost of scaling a single reasoner. There must exist a tipping point where cumulative inference cost exceeds the avoided training budget, and our conservative calculation places this far beyond typical academic inference volumes. Additionally, the requirement of simultaneously hosting multiple models leads to a fixed memory premium whose *relative* impact diminishes for larger base models. Finally, not all academic groups can access clusters with 80 high-end GPUs. From a systems and economic perspective, PROXY-THINKER offers a practical path to high reasoning quality with substantially lower upfront compute.

## D  LIMITATIONS

While PROXYTHINKER demonstrates strong empirical performance and practical scalability, several limitations remain. First, our method relies on access to both an RFT expert and an amateur model sharing the same vocabulary and preferably sharing the same model architecture. This requirement may limit applicability in settings where high-quality RFT experts are unavailable or where model architectures are not easily aligned. Existing cross-model alignment methods, such as CDM (Chen et al., 2025c), might help mitigate this issue. In addition, our current experiments focus primarily

---

[1]We deliberately ignore the modest dependence on sequence length and note that modern KV cache designs further reduce effective compute per token, so our estimate is conservative (i.e., an overestimate).

on visual reasoning benchmarks. The effectiveness of PROXYTHINKER in other domains, such as natural language-only reasoning or real-world embodied tasks, remains to be thoroughly explored.

## E   THE USE OF LARGE LANGUAGE MODELS (LLMS)

We employed large language models (LLMs) only for language refinement, such as correcting grammar and enhancing clarity. The LLM was not involved in generating ideas, conducting analysis, or contributing substantive content. All conceptual framing, methodological design, results, and interpretations were carried out solely by the authors.

