# OpenReview forum: "ProxyThinker: Test-Time Guidance through Small Visual Reasoners"
_ICLR.cc/2026/Conference — ICLR 2026 Poster_

### Official Review · Reviewer_rAEa · 2025-10-30

**Soundness:** 3
**Presentation:** 4
**Contribution:** 3
**Rating:** 6
**Confidence:** 4

**Summary:**

This paper introduces ProxyThinker, a test-time decoding method that transfers visual reasoning abilities from small RFT models to LVLM without any additional training. By applying the token-level logits difference between a small RFT "expert" and its non-RFT "amateur" counterpart to the large base model's output distribution, ProxyThinker effectively elicits "slow-thinking" behaviors such as self-verification and backtracking. The method achieves consistent improvements on challenging benchmarks, often matching or surpassing full RFT-trained models. Analyses show that ProxyThinker, not only enhances quantitative performance but also induces interpretable reasoning behaviors.

**Strengths:**

1. This paper presents clear motivation and method. The method is not complex but effective, which is easy to follow.

2. The system implementation is carefully engineered on top of vLLM, with tensor-parallel scheduling and asynchronous multi-model inference that achieve up to 38× speedup.

3. Experiments are thorough across five visual reasoning benchmarks and two model scales (32B/72B).  The results are consistent and reported with clear baselines and ablations.

**Weaknesses:**

1. The paper begins by arguing that RFT does not teach new knowledge but merely amplifies existing reasoning patterns.  However, ProxyThinker itself inherits exactly those amplified patterns without introducing new reasoning capabilities. The proposed issue is not well resolved.

2. The related-work section omits some concurrent or near-concurrent works on multimodal reasoning with RL or decoding-time control, such as

- Open Vision Reasoner: Transferring Linguistic Cognitive Behavior for Visual Reasoning (Wei et al., 2025)

- R1-ShareVL: Incentivizing Reasoning Capability of Multimodal Large Language Models via Share-GRPO (Yao et al., 2025)

- Semi-Off-Policy Reinforcement Learning for Vision-Language Slow-Thinking Reasoning (Shen et al., 2025)

3. While ProxyThinker improves on math- and spatial-reasoning benchmarks, gains on knowledge-intensive datasets such as MMMU-Pro are marginal or even negative with weaker experts.  This suggests the transferred signal mainly benefits structural reasoning rather than factual grounding.

4. ProxyThinker requires simultaneous execution of three large models: a base model plus two auxiliary 7B-scale models (expert and amateur).  This setup substantially increases both memory footprint and compute parallelism requirements, which could be prohibitive for typical deployment environments where even a single 32B LVLM may require multiple GPUs.

**Questions:**

See above weaknesses.

---

> ### Author Response · Authors · 2025-11-20
>
> We are grateful for your positive recognition of ProxyThinker and would love to address your concerns below.
>
> ---
>
> > W1 & W3. ProxyThinker inherits reasoning patterns but does not introduce new knowledge. This issue is not well solved. Marginal improvements on knowledge-intensive benchmarks suggest that the transferred signal primarily benefits structural reasoning rather than factual grounding.
>
> We thank the reviewer for raising this point. Our intention in Lines 143-149 is not to frame “no new knowledge” as a flaw of RFT, but to clarify **what RFT is actually doing**: it primarily reshapes and amplifies existing reasoning patterns of a strong base model rather than injecting new factual knowledge. And such characteristics motivates the design of applying logit delta in ProxyThinker. We consider the “no new knowledge” more like **a feature of RFT, not a drawback.**
>
> Thus, the observation that ProxyThinker does not create new knowledge or fundamentally new reasoning capabilities is **expected by design**, and consistent with our broader claim. Even when we apply RL directly (RFT experts) or indirectly (ProxyThinker), improvements on knowledge-intensive tasks are limited. Our contribution is to show that those reasoning gains can be transferred in a training-free manner, not to change the underlying knowledge profile of the base model.
>
> Regarding the marginal improvements on knowledge-intensive benchmarks, we believe **you could view ProxyThinker as a type of test-time distilling methods.** Distillation can usually only extract and propagate what is already present in the teacher. If the RFT expert itself does not substantially enhance knowledge-related competence, then a training-free distillation-like approach such as ProxyThinker has no source of new factual signal to learn from. For instance, VL-Rethinker-7B [1] Figure 7 reports a relative improvement of just 1.84% on MMMU-Pro, in stark contrast to its larger boosts on math datasets like MathVision, MathVista, and MathVerse (8.46%, 3.52%, and 1.98%, respectively). This suggests that even when we directly fine-tune a model with RL for better reasoning, the additional benefit on knowledge-intensive tasks remains limited.
>
> ---
>
> > W2. Concurrent / Near-Concurrent Works on multimodal reasoning with RL
>
> Thank you for suggesting those great works! They cover different aspects of visual reasoning transfer and variants of policy optimization methods. Please refer to Lines 495-498 for the updated discussion.
>
> ---
>
> >  W4. Memory and computation overhead.
>
> We agree that collaborative decoding with three models increases instantaneous memory and compute demand. We hope you could understand that this is an inherent property of decoding-time steering approaches such as DExperts and contrastive decoding, rather than a limitation specific to ProxyThinker.
>
> Our contribution is to make such steering practical at scale: we implement collaborative decoding in vLLM, and we schedule the base, expert, and amateur models asynchronously on separate tensor-parallel groups. This system's design keeps all GPUs highly utilized and minimizes idle time, leading to up to 38$\times$ higher throughput than prior open-source decoding-time steering baselines.
>
> As a result, the added latency is modest compared to the base model decoding time, and the *relative* memory overhead diminishes as the base model size increases. Appendix C provides a detailed discussion regarding the memory and computation overhead.
>
> ---
>
> We sincerely thank you again for your efforts in reviewing which helped a lot in improving our paper.
>
> References:
>
> [1] Wang, Haozhe, et al. "Vl-rethinker: Incentivizing self-reflection of vision-language models with reinforcement learning." *arXiv preprint arXiv:2504.08837* (2025).

---

> > ### Comment · Reviewer_rAEa · 2025-11-22
> > **Thanks for your response.**
> >
> > Thanks to the authors for the clear and comprehensive response. My concerns have been addressed, and the revisions improve the paper. I am upgrading my score.

---

> > > ### Author Response · Authors · 2025-11-22
> > > **Thank you for your positive feedback!**
> > >
> > > Dear Reviewer rAEa,
> > >
> > > Thank you for updating your evaluation of our work and we are glad your comments and our revisions put ProxyThinker into a better shape.

---

### Official Review · Reviewer_nfVY · 2025-10-31

**Soundness:** 3
**Presentation:** 3
**Contribution:** 3
**Rating:** 8
**Confidence:** 4

**Summary:**

This paper proposes PROXYTHINKER, a novel and training-free decoding strategy designed to elicit multi-step reasoning from large vision-language models without expensive training. The method operates at inference time, steering a large base model's generation by applying a guidance signal derived from the logit difference between a small, RFT expert model and its non-RFT amateur counterpart. Experiments compellingly demonstrate that this approach effectively transfers the reasoning skills, enabling a 32B model to achieve performance on complex visual reasoning benchmarks that is comparable to, and in some cases surpasses, its fully RFT-trained equivalent, showcasing a highly efficient path to enhancing model reasoning.

**Strengths:**

1. The paper is exceptionally well-written, clearly articulated, and easy to follow. The motivation is strong and well-grounded, directly addressing the significant and timely challenge of improving the scalability of reinforcement learning for large-scale VLMs.

2. The core idea is highly intuitive and logically sound. It builds upon the key insight that RFT methods like GRPO often do not introduce new external knowledge but rather reshape the model's output distribution to elicit a step-by-step reasoning process. The authors astutely hypothesize that if this capability is already latent within a large instruction-tuned model, it can be unlocked via a direct, inference-time decoding control mechanism. This work provides a feasible and effective solution that validates this hypothesis with impressive results.

3. This research could pioneer a new and more scalable paradigm for building powerful reasoning systems. It suggests a practical "division of labor," where intensive and costly reinforcement learning is focused on creating compact, specialized "expert" models. Larger, more general models could then be trained primarily with SFT and efficiently endowed with these specialized reasoning skills at inference time through interaction, paving a more sustainable path for model development.

**Weaknesses:**

1. Lack of Principled Analysis on Expert Model Selection: The paper's primary contribution relies on guidance from a small "expert" model, yet the criteria for selecting this expert seem somewhat ad-hoc. While the authors experiment with three public models chosen based on "differing training paradigms and data selection strategies" (line 259), the paper falls short of addressing a crucial question: What properties define an optimal expert for the PROXYTHINKER framework? A deeper investigation is needed into the essential characteristics an expert must possess. For instance, is it absolute accuracy, the diversity of reasoning paths, or the magnitude of the logit shift post-RFT? A more solid contribution would involve discussing these factors and perhaps even exploring how one might intentionally train a more effective, targeted expert model specifically for this guidance role. Such an analysis would significantly enhance the paper's impact and provide more concrete guidance for the community.

2. Limited Generalizability of Base Models: The empirical validation exclusively uses models from the Qwen2.5-VL series as the base model. While these are strong models, this narrow selection raises concerns about the generalizability of the proposed method. It is plausible that the success of PROXYTHINKER is partially contingent on the specific latent reasoning capabilities inherent in the Qwen architecture and its instruction-tuning process. The paper's claims would be substantially more robust if the authors demonstrated effectiveness on other diverse and widely-used LVLM families (e.g., InternVL / LLaVA-OV). Without such experiments, it remains an open question whether this is a universally applicable decoding technique or one that works particularly well for a specific class of models.

**Questions:**

See Weaknesses

---

> ### Author Response · Authors · 2025-11-20
>
> Thank you for your thoughtful comments and positive feedback on ProxyThinker! We are glad that you find ProxyThinker to be intuitive and logically sound.
>
> ---
>
> > W1. Analysis on Expert Model Choice.
>
> We agree that understanding what makes a good ProxyThinker expert is important, and we appreciate the opportunity to clarify our expert choices.
>
> Our main empirical finding is that the **overall reasoning strength of the expert, measured by its average accuracy across multiple benchmarks, is a practical and predictive indicator** of how beneficial it will be for ProxyThinker.
>
> | **Model**        | **Expert** | **Math Vista** | **Math Verse** | **Math Vision** | **MMMU Pro** | **R1-Bench** | **Δ** |
> | ---------------- | ---------- | -------------- | -------------- | --------------- | ------------ | ------------ | ----- |
> | Qwen2.5-VL-7B    | –          | 68.2           | 46.3           | 25.1            | 36.9†        | 32.1         | 0.0   |
> | OpenVLThinker-7B | –          | 70.2           | 47.9           | 25.3            | 39.4*        | 32.9*        | +1.4  |
> | ThinkLite-VL-7B  | –          | 75.1           | 50.7           | 32.9            | 41.1*        | 39.0*        | +6.0  |
> | VL-Rethinker-7B  | –          | 74.9           | 54.2           | 32.3            | 41.7         | 47.2*        | +8.3  |
>
> The above table lists the average improvement over the base model for different expert models. As you may have noticed, experts with **higher average multi-task accuracy** usually lead to consistently larger gains for ProxyThinker-72B in Table 1. Also, a clearly underperforming expert like VLAA-Thinker mentioned Appendix B.2 induces **negative transfer. Based on those observations, we provide two concrete practical guidelines for ProxyThinker users:**
>
> 1. Use the strongest available RFT expert in the target domain.
> 2. Assess multi-task reasoning quality rather than a single score. We find that the performance of a single benchmark can be noisy or unrepresentative. Average performance across multiple datasets, together with qualitative inspection of reasoning traces, is more informative about the expert’s suitability.
>
> ---
>
> > W2. Limited Generalizability of Base Models
>
> The availability of strong open-weight RFT experts drives our focus on Qwen2.5-VL. Recent multimodal RL / RFT works **build and ONLY build on Qwen2.5-VL backbones** and release their checkpoints as “reasoning experts” for visual reasoning. And since ProxyThinker requires access to strong, **open** RFT checkpoints, we have no choice but to also select **Qwen2.5-VL series expert models.**
>
> Although our main VLM experiments use Qwen2.5-VL, we *do* validate other base models in the **language-only** setting per reviewer cUEK’s suggestions, where we can easily obtain diverse experts in different backbones. Refer to **Appendix B.3** for details. In particular, using a *Llama3.1-8B-RL* expert to guide *Llama3.1-70B*, ProxyThinker improves Math500 from **22.2 -> 27.0** and GSM8K from **42.4 -> 52.9**, despite the expert itself being relatively weak. We hope this language-only setting could alleviate your concerns regarding limited generalizability.
>
> ---
>
> We hope our responses above could address your concerns. We are always happy to engage in discussions.

---

### Official Review · Reviewer_cUEK · 2025-10-31

**Soundness:** 3
**Presentation:** 3
**Contribution:** 2
**Rating:** 4
**Confidence:** 3

**Summary:**

The paper proposes PROXYTHINKER, a training-free decoding method that transfers “slow-thinking” visual-reasoning behaviors from a small RFT-trained expert VLM to a larger base VLM by adding a logits delta at every generation step. The approach consistently improves Pass@1 on visual math and multi-disciplinary benchmarks and sometimes approaches or even surpasses a fully RFT-trained large model. The authors further implement a multi-model scheduling scheme in vLLM to reduce the overhead of running three models at inference.

**Strengths:**

# Strengths

1. Practical inference-time approach: The method requires no additional training of the large model, addressing the high cost of RFT for VLMs. It is simple to implement (just logit arithmetic) and can leverage existing small RFT models.

2. Empirical gains on visual reasoning tasks: ProxyThinker consistently improves accuracy on spatial and math reasoning benchmarks. In many cases the base VLM closes most of the gap to a fully RFT-trained model. For example, applying a small visual “OpenVLThinker-7B” expert to Qwen2.5-32B raises MathVision accuracy from 38.4% to 40.8%, slightly exceeding the 40.5% achieved by the full-scale RFT model.

**Weaknesses:**

# Weaknesses

1. Overall, the novelty of the proposed method is under par. ProxyThinker closely mirrors existing logit-guidance techniques. The existence of current methods, like DExperts, Proxy-Tuning and DoLa, makes the ProxyThinker not novel enough.

2. Weak justification for VLM-specific focus. The authors motivate the work by the expense of RFT on large VLMs, but they do not identify any modality-specific challenge that makes ProxyThinker inherently necessary for vision. The technique appears generic and could equally apply to language-only reasoning tasks.

3. Although the gains are statistically significant, they are small (often only a few percentage points). Crucially, ProxyThinker requires simultaneous inference of multiple models (base + expert + amateur), which increases serving complexity, latency, and cost; in many settings this overhead may not be justified by the marginal improvements over fully RFT-trained baselines.

**Questions:**

# Questions:

1. Did you try to apply the proposed method on language-only reasoning tasks? What are the challenges and performance?

2. Did you observe any negative side effects (e.g. increased repetition, verbosity, or unnatural reasoning) from adding the logit shift?

3. On MMMU/MMMU-Pro the gains are negligible. Could you provide a detailed diagnosis?

---

> ### Author Response · Authors · 2025-11-20
> **Response (Part 1/2)**
>
> Thanks for your constructive feedback! We would love to address your concerns as below.
>
> ---
>
> > W1. Novelty Compared to Existing Decoding-Time Steering Methods (DExperts, Proxy-Tuning, etc.)
>
> We appreciate the reviewer’s observation that our method belongs to the family of logit-guidance approaches, and we agree that it is important to clarify how ProxyThinker goes beyond prior work such as DExperts, Proxy-Tuning, and DoLa.
>
> 1. **Different goal: transferring reasoning**, not attributes: DExperts and Proxy-Tuning are designed to steer generation toward **attributes** such as low toxicity, sentiment, or style. Their focus is on controlling surface properties of text given a fixed task, which limits generalization, for example, a toxicity expert in ProxyTuning cannot be reused for QA. In contrast, ProxyThinker explicitly targets the **transfer of** **procedural reasoning ability** from a small RFT expert into a larger base model.
> 2. **Domain-agnostic reasoning transfer, not domain-specific control**: In ProxyThinker, the expert captures a *general reasoning prior*: in our experiments, even when the expert is trained only on Physics, applying ProxyThinker yields consistent gains across Math, Physics, Chemistry, and Biology. This cross-domain generalization (Appendix B.1) is precisely what we show by domain-agnostic reasoning transfer, and to our knowledge is not demonstrated in prior decoding-time steering work.
> 3. **System-level contribution and scalability: **ProxyThinker is implemented as a collaborative decoding framework on top of a modern high-performance inference engine, with communication-layer optimizations that enable practical, large-scale deployment. In our measurements, a naive HuggingFace-style implementation of prior methods required ~5 hours on 8 GPUs to answer 304 MathVista samples, while ProxyThinker completed the same workload in under 10 minutes.
>
> In summary, while ProxyThinker builds on the general idea of manipulating logits at decoding time, its **target (reasoning transfer),  cross-domain behavior**, and **system-level implementation** differ substantially from existing works.
>
> ---
>
> > W2 & Q1. Weak justification for VLM-specific focus. Can ProxyThinker be applied in a language-only setting?
>
> Thank you for proposing the language-only experimental settings. We agree that ProxyThinker design is not restricted to the visual modality. We now add additional experiments on *language-only* reasoning benchmarks (Math500, AIME’24, GSM8K), **summarized in Appendix B.3.** These experiments show that ProxyThinker consistently improves the performance of strong base LLMs and can approach or even match the performance of much more expensive RL-fine-tuned experts, confirming the generality of our method beyond VLMs and echoing the similar conclusion in our submission. We believe those results can provide further evidence of the effectiveness and will organize them into the main paper upon the review decision.
>
> ---
>
> > W3. Small improvements might not justify the additional inference overhead.
>
> We would like to address your concern W3 in two folds.
>
> **A. Small Improvements**
>
> We agree that Table 1 can show only small gains. As mentioned in L223, the results were reported with a fixed α = 0.5, chosen arbitrarily. For completeness, we now provide the best results found through hyperparameter search. Those results are also reported in curves in Figure 4.
>
> | **Source**               | **Model**        | **ProxyThinker Expert** | **MathVista**      | **MathVerse**      | **MathVision**     |
> | ------------------------ | ---------------- | ----------------------- | ------------------ | ------------------ | ------------------ |
> | Base Model Performance   | Qwen2.5-VL-32B   | -                       | 74.7               | 53.8               | 38.4               |
> | Table 1                  | Qwen2.5-VL-32B   | VL-Rethinker-7B         | 78.1 (+3.4)        | 55.1 (+1.3)        | 39.2 (+0.8)        |
> | Best $\alpha$              | Qwen2.5-VL-32B   | VL-Rethinker-7B         | 78.6 (+3.9) ($\alpha$=0.3) | 57.2 (+3.4) ($\alpha$=1.2) | 40.3 (+1.9) ($\alpha$=0.1) |
> | Full-scale RFT reference | VL-Rethinker-32B | -                       | 78.8 (+4.1)        | 56.9 (+3.1)        | 40.5 (+2.1)        |
>
> We find that with a well-chosen α, the 32B base model can match or even surpass the performance of a strong 32B RFT expert. To maintain the generality **of ProxyThinker**, we intentionally avoided reporting cherry-picked optimal results in Table 1. In practice, one could use a held-out evaluation to determine those choices when applied to a specific reasoning domain.
>
> (To be continued)

---

> > ### Author Response · Authors · 2025-11-20
> >
> > **B. Additional Overhead**
> >
> > The need to run multiple models is a general property of decoding-time steering methods (e.g., DExperts, contrastive decoding), and not unique to ProxyThinker. We therefore optimized the systems side aggressively: we implement collaborative decoding in vLLM, tensor parallelism and continuous batching, and schedule the base, expert, and amateur models asynchronously on separate tensor-parallel groups. As shown in our implementation section and Fig. 3, this minimizes GPU idle time and yields up to **38× throughput** over earlier open-source decoding-time steering baselines. In practice, the added latency is modest relative to the base model’s decoding time, while memory overhead becomes relatively smaller as the base model size grows. Appendix C provides a complete discussion of additional overhead.
> >
> > ---
> >
> > > Q2. Side effects of logit shift
> >
> > Thank you for bringing this point up! In light of your response, we design similar LLM-as-a-judge strategy as indicated in Lines 402-410 to classify reasoning trajectories of the large base models and ProxyThinker with the following prompt derived from [1].
> >
> > ```markdown
> > # Task Description
> >
> > You will be provided with text from the internet.
> >
> > Classify the text to one of the following categories: Repetition, Verbosity, Unnatural Reasoning. If none are present, classify it as "Acceptable."
> >
> > # Evaluation Criteria
> >
> > Analyze the provided reasoning process and determine if it exhibits any of the following characteristics. If multiple characteristics are present, select the one that is most dominant.
> >
> > 1. Repetition
> >
> > - Definition: The text unnecessarily restates the same idea, key phrase, or piece of information multiple times. This is distinct from standard introductory/concluding summary; it refers to inefficient, redundant self-reinforcement within the core body of the reasoning.
> >
> > 2. Verbosity
> >
> > - Definition: The text uses significantly more words than necessary to convey its message. It includes excessive filler words, overly complex sentence structures, or lengthy, indirect explanations that could be made concisely. The reasoning is correct but padded.
> >
> > 3. Unnatural Reasoning
> >
> > - Definition: The logical flow is stilted, non-human, or illogical. This includes sudden, unexplained leaps in the argument, circular logic, awkward transitions, or phrasing that suggests the reasoning was generated by a system that lacks human intuition.
> >
> > # Task Format
> >
> > Format your response in markdown as follows:
> >
> > ## Thoughts
> >
> > [Brief description describing what behavior was noticed]
> >
> > ## Classification
> >
> > [One of: Increased Repetition, Verbosity, Unnatural Reasoning, Acceptable]
> >
> > # Text to evaluate
> >
> > {response}
> >
> > # Response
> > ```
> >
> > | **Model**        | Repetition | Verbosity | Unnatural | Acceptable |
> > | ---------------- | ---------- | --------- | --------- | ---------- |
> > | Qwen2.5-VL-32B   | 562        | 363       | 39        | 2076       |
> > | ProxyThinker-32B | 602        | 394       | 17        | 2027       |
> >
> >
> > Note that the repetition defined in the instruction prompt is much stricter, so the absolute number is not indicative. However, the relative numbers indicate that **ProxyThinker does not introduce statistically significant side effects to the reasoning process.**
> >
> > ---
> >
> > > Q3. The reason behind MMMU-Pro negligible improvements.
> >
> > We acknowledge that ProxyThinker’s negligible on knowledge-intensive tasks is a limitation, however, such limitation is inherited from the *reinforcement fine-tuning (RFT) experts themselves*, rather than being specific to our test-time guidance mechanism.
> >
> > On knowledge-heavy benchmarks like MMMU/Pro, existing RFT-based reasoning models also show only **small relative gains** compared to their strong baselines. For example, VL-Rethinker-7B [1] Figure 7 reports a relative improvement of **1.84% on MMMU-Pro**, which is much smaller than its gains on math datasets such as MathVision, MathVista, and MathVerse (8.46%, 3.52%, 1.98% respectively).
> >
> > This indicates that even when we directly fine-tune a model with RL for reasoning, the benefit on knowledge-intensive tasks is minimal. Therefore, a training-free method like ProxyThinker, which reuses such experts, cannot create large gains well because ProxyThinker is guided by signals from an RFT expert.
> >
> > ---
> >
> > We sincerely appreciate your review in helping improve our paper and hope our response will be satisfactory. Please feel free to leave us more comments, and we will be happy to engage in discussions.
> >
> >
> >
> > References:
> >
> > [1] Gandhi, Kanishk, et al. "Cognitive behaviors that enable self-improving reasoners, or, four habits of highly effective stars." *arXiv preprint arXiv:2503.01307* (2025).

---

### Official Review · Reviewer_x1re · 2025-10-31

**Soundness:** 3
**Presentation:** 3
**Contribution:** 3
**Rating:** 6
**Confidence:** 4

**Summary:**

The paper proposes a training-free method to improve the visual reasoning of large vision-language models (LVLMs).
The core problem it addresses is the immense computational cost of full-scale Reinforcement Fine-Tuning (RFT) on large models.
The paper proposes to use logit delta to transfer the "slow-thinking" reasoning behaviors (like self-correction) from the small expert to the large base,

**Strengths:**

1. The paper spot on a valuable problem of training cost of RL.

2. The paper delivers a surprising and useful finding that the reasoning behavior can be transfered from small expert model to large base model, alleviating the burden of training cost.

3. The authors have thoroughly addressed the practical viability of using three models at inference. By leveraging vLLM and optimized tensor parallelism, they demonstrate a ~38x speedup over a naive implementation. Their system adds only a minor latency overhead (~11%) compared to running a single large model, making it a genuinely practical and economical solution for deploying advanced reasoning capabilities.

**Weaknesses:**

1. A Significant Trade-off in Reasoning Diversity (Pass@k): The paper's claim of "striking a balance" in reasoning exploration (Sec 4.2) is an oversimplification. The data in Figure 5 clearly shows that while Pass@1 performance (greedy decoding) is improved, the Pass@k performance for $k>4$ drops below that of the unguided base model. This suggests the guidance narrows the large model's reasoning diversity, forcing it down the single "slow-thinking" path favored by the small expert. This trade-off—improving the single best answer at the cost of exploratory reasoning capacity—is a major weakness that is not sufficiently discussed.

2. Unexplored Mechanism of Action: The paper's central hypothesis—that the logit delta $(z_{\text{expert}} - z_{\text{Amateur}})$ isolates a scalable, model-agnostic "reasoning vector"—is demonstrated that it works but not why it works. The ablation in Figure 4 proves that subtracting the amateur model is critical, but the paper provides no deeper analysis into the nature of this delta. Is it merely up-weighting a shared vocabulary of "reasoning tokens" (e.g., "Wait," "let's check"), or is it manipulating a deeper, abstract representation of the reasoning process? The simplicity of the mechanism makes this lack of explanation a significant gap.

3. The method's effectiveness is strictly bounded by the small expert's capabilities.
- As admitted in Appendix A.4, the method fails on knowledge-intensive tasks (like MMMU) where the small expert lacks the requisite knowledge. It cannot transfer knowledge, only reasoning patterns, and a correct pattern applied to incorrect knowledge still fails.

- As shown in Appendix B.2, a low-quality expert results in negative transfer, actively harming the large model's performance. This implies the method is not a universal booster; it requires a small expert that is both knowledge-sufficient for the domain and has been RFT-trained to a high-quality "floor."

**Questions:**

1. Could you elaborate on the Pass@k trade-off? Do you view the loss of reasoning diversity (worse Pass@k performance) as an acceptable price for the gain in Pass@1 performance? Does this imply RFT-style reasoning (at least as captured by this method) inherently prunes other valid reasoning paths?

2. Could you elaborate on why a simple logic delta could work so well, and what might be the premise of the success achieved from applying this logit delta? Following on Weakness #2, what is in the delta $(z_{\text{expert}} - z_{\text{Amateur}})$? Have you analyzed its composition? For instance, does the delta vector's direction change over a reasoning trace, perhaps pushing toward "sub-goal" tokens at the start and "verification" tokens near the end? A temporal analysis of the delta's top-k components would be fascinating.

3. The knowledge bottleneck (Appendix A.4) is a key limitation. How do you see this method performing on tasks that are a tight mix of reasoning and knowledge? Is there a risk that the small, knowledge-poor expert could misguide the large, knowledge-rich base model into a factually incorrect but "well-reasoned" answer?

4. Given that a poor expert causes "negative transfer" (Table 5), how should one practically determine if a small expert is "good enough" to be a PROXYTHINKER? Its own standalone accuracy seems unreliable (e.g., the OpenVLThinker-7B had only 25.3% on MathVision but gave a positive boost). Is there a "floor" of capability or a metric (beyond final accuracy) that predicts positive transfer?

---

> ### Author Response · Authors · 2025-11-20
> **Response (Part 1/2)**
>
> We appreciate your positive feedback on ProxyThinker. We are glad that you find ProxyThinker to be a practical solution to transfer visual reasoning capabilities without training. Below, we present our response to your remaining concerns.
>
> ---
>
> > **W1 & Q1. ProxyThinker is improving the single best answer at the cost of exploratory reasoning capacity. Is the worse Pass@k performance an acceptable price for the gain in Pass@1 performance?**
>
> We thank the reviewer for bringing this important point to our attention. We agree that the phrase “striking a balance” may read as overly strong if interpreted as “**no loss** of reasoning diversity” compared to the base model.
>
> We believe the observation in Section 4.2 can be summarized into the following empirical findings:
>
> 1. **ProxyThinker improves Pass@1** over the base model and over the small RFT expert.
> 2. **At larger k**, the base model eventually achieves higher Pass@k than both the RFT expert and ProxyThinker, echoing the Reinforcement Learning with Verifiable Reward (RLVR) behavior discussed in Yue et al, that current RLVR methods simply elicits the reasoning patterns already in the base model.
> 3. ProxyThinker’s Pass@k curve **lies between the base and the RFT expert**, i.e., it does not collapse as much as full RFT, but it also does not preserve the full exploration capacity of the base model.
>
> Thus, ProxyThinker *partially* inherits the RLVR trade-off: it improves sample efficiency at small $k$ by concentrating probability mass on slow-thinking trajectories, but this necessarily reduces coverage of low-probability reasoning paths that contribute to Pass@k at very large $k$. While this might not be optimal, it is expected because ProxyThinker is a training-free method that modifies the decoding dynamics **without any external signals other than the small expert model.**
>
> For your question, we think the answer is yes. Within the standard RLVR setting we target, we consider the observed Pass@k drop to be an acceptable and in fact *inherent* price for the Pass@1 gain. This phenomenon is not unique to ProxyThinker, but rather a well-documented property of nearly all RLVR-style post-training models. In fact, what our results in Figure 5 show is that **ProxyThinker somewhat** **mitigates** **this weakness compared to a fully RL-trained expert.**
>
> In light of your comment, we revised the statement in the submission to more accurately reflect the observed behaviours.
>
> ---
>
> > **W2. Why does ProxyThinker work? What’s in the delta? For instance, does the delta vector's direction change over a reasoning trace, perhaps pushing toward "sub-goal" tokens at the start and "verification" tokens near the end? A temporal analysis of the delta's top-k components would be fascinating.**
>
> We appreciate the reviewer’s request for an explanation of why ProxyThinker logits delta works beyond the empirical gains. In response to this, we added an additional Section 5 to incorporate the temporal analysis of the logit delta, which activates self-reflection in the decoding process. To summarize:
>
> 1. At early steps, the bar plots of ∣Δ∣ are small and balanced, and the top-3 Δ tokens include some non-ASCII artifacts because of the byte-level tokenization. This indicates that the expert and amateur logit distributions are nearly aligned, so ProxyThinker leaves decoding logits unchanged at these positions.
> 2. At late steps, Δ becomes structured as the largest positive delta corresponds to tokens like *Wait.* This shows that Δ actively steers the large model toward inserting a pause and reconsideration with self-reflection attempts and finally results in a self-verification in the reasoning process.
>
> (To be continued)

---

> ### Author Response · Authors · 2025-11-20
> **Response Part (2/2)**
>
> > **W3.1 & Q3. ProxyThinker does not improve on knowledge-intensive tasks. Is there a risk that the small, knowledge-poor expert could misguide the large, knowledge-rich base model into a factually incorrect but "well-reasoned" answer?**
>
> We acknowledge that ProxyThinker not improving on knowledge-intensive tasks is a limitation, however, such limitation is inherited from the *reinforcement fine-tuning (RFT) experts themselves*, rather than being specific to our test-time guidance mechanism.
>
> On knowledge-heavy benchmarks like MMMU/Pro, existing RFT-based reasoning models also show only **small relative gains** compared to their strong baselines. For example, VL-Rethinker-7B [1] Figure 7 reports a relative improvement of **1.84% on MMMU-Pro**, which is much smaller than its gains on math datasets such as MathVision, MathVista, and MathVerse (8.46%, 3.52%, 1.98% respectively).
>
> This indicates that even when we directly fine-tune a model with RL for reasoning, the benefit on knowledge-intensive tasks is minimal. Therefore, a training-free method like ProxyThinker, which reuses such experts, cannot create large gains as well.
>
> Regarding the risk of “misguiding” the base model, **thanks to your insights in temporal analysis,** we found that ProxyThinker operates at the **logit-delta level** and mostly modifies **reasoning-control** **dimensions rather than content tokens.** The temporal analysis in Section 5 (Fig. 6 in the paper) shows that the largest components of the logit delta $\Delta_t$ correspond to meta-cognitive control tokens such as *Wait*, *Verify*, and *Identify*, while content tokens remain largely aligned between base and expert when their predictions agree. This suggests that ProxyThinker mainly steers *how* the model reasons (e.g., whether to reflect or verify), rather than overwriting the base model’s factual knowledge.
>
> Empirically, on knowledge-intensive benchmarks we observe that ProxyThinker’s performance is close to the base model and does not degrade it, while on reasoning-centric tasks we consistently improve over both the base and the small expert. This pattern is consistent with recent findings that RFT tends to **preserve general knowledge [2]**, rather than harming it.
>
> Overall, the risk of such a misguidance is minimal both theoretically and empirically.
>
> ---
>
> > **W3.2 & Q4. Effectiveness is strictly bound by the small expert. Then how do we define whether an expert is good enough to improve the large base model?**
>
> We thank the reviewer for highlighting the dependence of ProxyThinker on the quality of the small expert. Our experiments, however, suggest that *overall reasoning strength* of the expert, measured by its average accuracy across multiple benchmarks, is a reasonably good practical indicator of whether it will help.
>
> | **Model**     | **Math Vista** | **Math Verse** | **Math Vision** | **Avg $\Delta$** |
> | ------------- | -------------- | -------------- | --------------- | ---------------- |
> | VLAA-Thinking | 61.0           | 36.4           | 24.4            | 0.0              |
> | OpenVLThinker | 70.2           | 47.9           | 25.3            | +7.2             |
> | ThinkLite-VL  | 75.1           | 50.7           | 32.9            | +12.3            |
> | VL-Rethinker  | 74.9           | 54.2           | 32.3            | +13.2            |
>
> We draw two practical guidelines for selecting a “good enough” ProxyThinker expert:
>
> 1. **Use the strongest available RFT expert in the target domain.** In our experiments, every expert that is competitive or better than VLAA-Thinking-7B on average across benchmarks yields non-negative gains, while the clearly underperforming expert causes negative transfer. This supports our statement in the paper that “the quality of the RFT expert generally determines the degree of PROXYTHINKER improvement.”
>
> 2. **Evaluate** **multi-task** **reasoning quality rather than a single score.** A single benchmark (e.g., MathVision) can be noisy or not fully representative. The average performance and qualitative structure of the expert’s reasoning paths are more predictive of positive transfer.
>
> ---
>
> We truly appreciate your time and constructive feedback. Should you have any additional questions or suggestions, we would be happy to provide further clarification.
>
> References:
>
> [1] Wang, Haozhe, et al. "Vl-rethinker: Incentivizing self-reflection of vision-language models with reinforcement learning." *arXiv preprint arXiv:2504.08837* (2025).
>
> [2] Zhang, Zhihao, et al. "Why Reinforcement Fine-Tuning Enables MLLMs Preserve Prior Knowledge Better: A Data Perspective." *arXiv preprint arXiv:2506.23508* (2025).

---

### Author Response · Authors · 2025-11-28
**Summary of Rebuttal Revisions and General Response**

Dear Reviewers and ACs,

Thank you for taking the time to review our paper! We are glad that reviewers recognized the strengths of ProxyThinker, and three of the four reviewers gave positive initial feedback. While it was a pity that we were unable to further discuss the feedback with the reviewers due to the recent incident, we would love to summarize the key points of feedback and the revisions we have made, with the differences **highlighted in BLUE** in the updated submission.

---
> **Reasoning diversity and Pass@k trade-off (x1re, cUEK)**

We explicitly acknowledge that ProxyThinker inherits the RLVR trade-off. We removed the overly strong “striking a balance” phrasing and **positioned the Pass@k behavior as expected and acceptable within the RLVR setting**, noting that ProxyThinker partially mitigates diversity collapse compared to fully RFT-trained experts.

---
> **New temporal analysis of the logit delta (x1re)**

**We added a new Section 5** with a temporal analysis of Δ over the reasoning trace. This shows that:

- Early steps: Δ is small and unstructured, indicating close alignment between expert and amateur logits, so ProxyThinker leaves the base logits largely unchanged.

- Later steps: Δ becomes structured and dominated by meta-cognitive control tokens (e.g., Wait, Verify), steering the large model to pause, reflect, and self-verify.

This supports the view that ProxyThinker primarily acts on reasoning-control dimensions, rather than directly overwriting content tokens.

---
> **Clarifying knowledge limitations and “misguidance” risk (x1re, cUEK, rAEa)**

We emphasize that the **limited gains on knowledge-intensive tasks are inherited from RFT experts themselves**, which already provide only marginal improvements on benchmarks like MMMU-Pro. We position this as a feature, not a drawback of RFT. It amplifies reasoning patterns more than it injects new knowledge.

Empirically, ProxyThinker does not degrade the base model on knowledge-heavy tasks, while yielding clear gains on reasoning-centric ones. Combined with the temporal analysis showing Δ focuses on control tokens, this suggests that the risk of being misled into "well-reasoned but factually incorrect" answers is minimal in practice.

---

> **Generalizing beyond VLMs and Qwen backbones (cUEK, nfVY)**

To demonstrate broader applicability, **we added language-only experiments in Appendix B.3.** Those experiments show that ProxyThinker works beyond VLMs and Qwen-series base models.

---
We hope these revisions and clarifications address the shared concerns while strengthening the overall narrative of ProxyThinker as a scalable test-time steering method for visual reasoning.

---

### Meta-Review · Area_Chair_EAoy · 2026-01-08

**Summary:**

the concerns were prior logit-steering based method, and the servicing hassle due to "three models at once" setup. Reviewer also pushed on two technical risks: Pass@k drops (less diversity) and the lack of a clear story for what the logit delta is doing. Multiple reviewers flagged limits on knowledge-heavy tasks (MMMU/MMMU-Pro) plus possible "well-reasoned but wrong" failure modes, and dependence on picking a strong enough expert. Generalization beyond Qwen2.5-VL backbones was another worry. But overall it's an interesting paper and received positive reviews, especially after the rebuttal.

**Reviewer Concerns:**

The rebuttal did a solid job: it openly owns the Pass@k trade-off, tones down overclaims, and adds a temporal delta analysis showing late-step steering is dominated by control tokens like "wait". It also adds language-only results and discusses side effects, plus better positioning on MMMU (limited because experts are limited too) and some practical guidance for choosing experts. What’s still not fully addressed: broad VLM-backbone generalization beyond qwen, and a truly principled expert-selection rule (right now it’s still mostly heuristic).

**Reviewer Scores:**

x1re: likely stays at 6, since the rebuttal answered their main questions.
cUEK: probably stays at 4.
nfVY: likely stays at 8 since the rebuttal reinforces.
rAEa basically confirmed by their comment saying concerns were addressed and they’re upgrading.

---

### Decision · Program_Chairs · 2026-01-26

Accept (Poster)